

# Impact of high resolution on Arctic Ocean simulations in Ocean Model Intercomparison Project phase 2 (OMIP-2)

Qiang Wang[1], Qi Shu[2,3], Alexandra Bozec[4], Eric P. Chassignet[4], Pier Giuseppe Fogli[5], Baylor Fox-Kemper[6], Andy McC. Hogg[7], Doroteaciro Iovino[5], Andrew E. Kiss[7], Nikolay Koldunov[1], Julien Le Sommer[8], Yiwen Li[9], Pengfei Lin[9], Hailong Liu[9], Igor Polyakov[10,11], Patrick Scholz[1], Dmitry Sidorenko[1], Shizhu Wang[2,3], and Xiaobiao Xu[4]

[1]Alfred Wegener Institute Helmholtz Centre for Polar and Marine Research (AWI), Bremerhaven, Germany
[2]First Institute of Oceanography, Key Laboratory of Marine Science and Numerical Modeling, Ministry of Natural Resources, Qingdao, China
[3]Shandong Key Laboratory of Marine Science and Numerical Modeling, Qingdao, China
[4]Center for Ocean–Atmospheric Prediction Studies, Florida State University, Tallahassee, FL, USA
[5]Ocean Modeling and Data Assimilation Division, Fondazione Centro Euro-Mediterraneo sui Cambiamenti Climatici (CMCC), Bologna, Italy
[6]Department of Earth, Environmental, and Planetary Sciences, Brown University, Providence, RI, USA
[7]Research School of Earth Sciences and ARC Centre of Excellence for Climate Extremes, Australian National University, Canberra, Australia
[8]Univ. Grenoble Alpes, CNRS, IRD, Grenoble INP, INRAE, IGE, Grenoble, France
[9]State Key Laboratory of Numerical Modeling for Atmospheric Sciences and Geophysical Fluid Dynamics, Institute of Atmospheric Physics, Chinese Academy of Sciences, Beijing, China
[10]International Arctic Research Center and College of Natural Science and Mathematics, University of Alaska Fairbanks, Alaska, USA
[11]Finnish Meteorological Institute, Helsinki, Finland

**Correspondence:** Qiang Wang (Qiang.Wang@awi.de)

**Abstract.** This study evaluates the impact of increasing resolution on Arctic Ocean simulations using five pairs of matched low- and high-resolution models within the OMIP-2 framework. The primary objective is to assess whether higher resolution can mitigate typical biases observed in low-resolution models and improve the representation of key climate-relevant variables. We reveal that increasing horizontal resolution contributes to a reduction in biases in mean temperature and salinity, and improves

the simulation of the Atlantic Water layer and its decadal warming events. Higher resolution also leads to improved agreement with observed surface mixed layer depth, cold halocline base depth and Arctic gateway transports. However, the simulation of the mean state and temporal changes in Arctic freshwater content does not show improvement with increased resolution. While the use of higher resolution demonstrates positive outcomes for certain variables, it is crucial to recognize that model numerics and parameterizations also play a significant role in achieving faithful simulations. Overall, higher resolution shows promise

in improving the simulation of key Arctic Ocean features and processes, but comprehensive model development is required to achieve more accurate representations across all climate-relevant variables.



## 1 Introduction

The Arctic is undergoing the most drastic anthropogenic changes on Earth, with the near-surface atmosphere warming two
to four times faster than the global average (known as Arctic Atmosphere Amplification; Holland and Bitz, 2003; Serreze

and Barry, 2011), the subsurface ocean warming about two times faster than the global average (known as Arctic Ocean
Amplification; Shu et al., 2022), and a significant retreat in sea ice extent, thickness, and volume (Kwok, 2018; Stroeve and
Notz, 2018; Fox-Kemper et al., 2021). The Arctic Ocean is connected to the global ocean through a few gateways (see Fig.
1). It receives ocean heat from the North Atlantic and North Pacific Oceans, and exports freshwater to the North Atlantic
Ocean. The ocean heat convergence into the Arctic Ocean and the hydrological cycle are expected to continue intensifying in

a warming climate (Wang et al., 2023). Numerical models play a crucial role in understanding the drivers and consequences
of these changes and predicting the future evolution of the climate. However, the accuracy of these models in representing the
different components of the Earth system and their interactions can influence our understanding and prediction (Lique et al.,
2016).

Past model intercomparison studies have revealed significant biases and spreads among ocean general circulation models in

simulating the hydrography, stratification, and gateway transports of the Arctic Ocean. In the Arctic Ocean Model Intercom-
parison Project (AOMIP), it was identified that a typical issue among models was an overly thick and deep Atlantic Water
layer in the Arctic Ocean (Holloway et al., 2007; Karcher et al., 2007), with numerical mixing suggested as the main cause
(Holloway et al., 2007). In the subsequent Coordinated Ocean-ice Reference Experiments phase II project (CORE-II; Griffies
et al., 2009), it was shown that the ocean general circulation models used in Coupled Model Intercomparison Project phase 5

(CMIP5) still struggled with the same issue a decade later (Ilicak et al., 2016). Furthermore, forced simulations of ocean mod-
els used in CMIP6 did not demonstrate significant improvements in representing the Atlantic Water layer in the Arctic Ocean
and exhibited large spreads in simulated basin mean temperatures (Shu et al., 2023). The model spreads in the temperature
of the Atlantic Water layer were suggested to be mainly due to differences in the simulated Atlantic Water inflows among the
models (Ilicak et al., 2016; Shu et al., 2023). The two generations of ocean models used in CMIP5 and CMIP6 also share other

common issues, including salinity biases in the halocline, overestimations of liquid freshwater content, and substantial spreads
in ocean volume, heat, and freshwater transports in Arctic gateways (Wang et al., 2016a; Ilicak et al., 2016; Shu et al., 2023).
The biases identified in forced ocean model simulations were also inherited and sometimes exacerbated in coupled climate
models of both CMIP5 (Shu et al., 2018, 2019) and CMIP6 (Zanowski et al., 2021; Khosravi et al., 2022; Wang et al., 2022b;
Muilwijk et al., 2023; Heuzé et al., 2023). It was found that ocean models usually perform better in representing the temporal

variability of Arctic gateway transports compared to their mean states (Wang et al., 2016a; Shu et al., 2023).

Higher model resolutions have been found to improve certain aspects of Arctic Ocean simulations. The narrowness of the
straits in the Canadian Arctic Archipelago makes it challenging to adequately represent the throughflow with the horizontal
resolutions typically used in CMIP models. As a result, there are significant model spreads within the ocean models used in
CMIP5 and CMIP6 in simulating the volume transport through the Davis Strait (Wang et al., 2016a; Shu et al., 2023). The

same issue is present even in ocean models dedicated for Arctic Ocean research (Jahn et al., 2012; Aksenov et al., 2016).



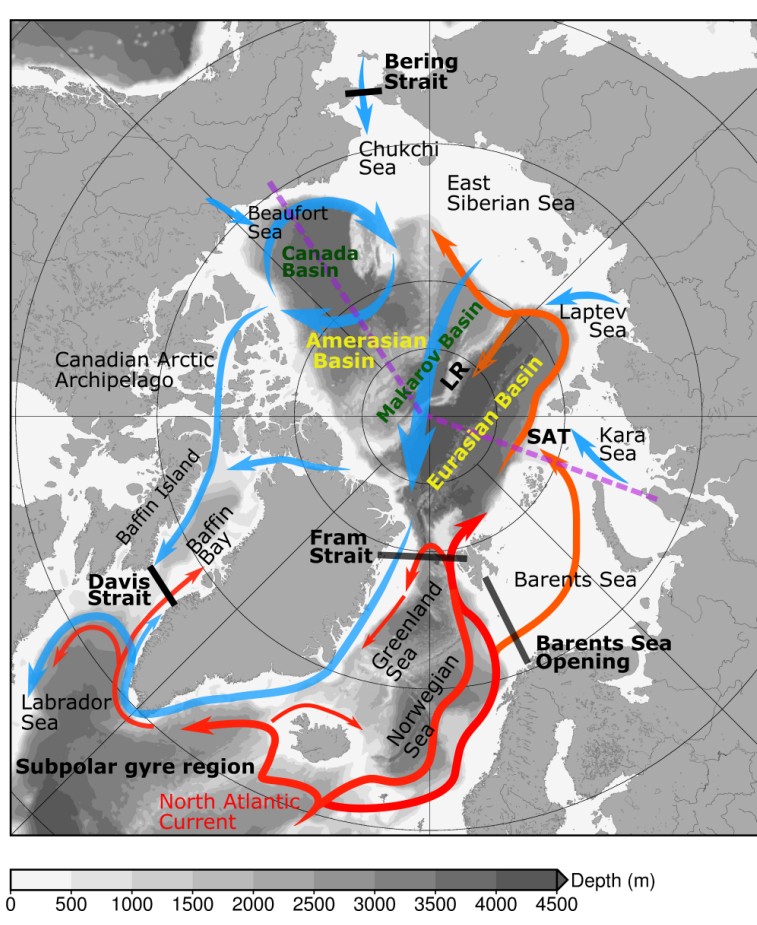

**Figure 1.** Schematic of pan-Arctic Ocean circulations. Blue arrows denote the circulations of low-salinity water, and red arrows denote the circulations of Atlantic Water. The background gray color in the ocean denotes bottom bathymetry. The four black lines denote the Arctic gateways of the Bering Strait, Davis Strait, Fram Strait, and Barents Sea Opening. The dashed magenta lines indicate the location of the transect shown in Fig. 6. LR and SAT are Lomonosov Ridge and St. Anna Trough, respectively.





However, when the horizontal resolution is increased to approximately 4 km, a forced ocean model simulation can more accurately reproduce the Canadian Arctic Archipelago throughflow (Wekerle et al., 2013). Low resolution was identified as one of the primary causes for the underestimation of Atlantic ocean heat transport to the Arctic Ocean in coupled climate models (Docquier et al., 2019). By utilizing resolutions that resolve eddies (approximately 1 km near Fram Strait), the transport of

Atlantic Water through the Fram Strait can be reasonably reproduced (Wekerle et al., 2017). Furthermore, the model bias of an overly thick Atlantic Water layer in the Arctic Ocean, persistently present in CMIP ocean models, can be significantly reduced by employing a horizontal resolution of around 4 km (Wang et al., 2018).

Within the framework of the Ocean Model Intercomparison Project phase 2 (OMIP-2, Griffies et al. (2016)), Chassignet et al. (2020) investigated the impact of horizontal resolution on global climate-relevant variables in four pairs of matched low- and

high-resolution ocean-sea ice simulations. They found that typical biases observed in low-resolution simulations, such as those related to the position, strength, and variability of western boundary currents, equatorial currents, and the Antarctic Circumpolar Current (identified in previous research by Tsujino et al., 2020), can be significantly improved in high-resolution models. However, the improvements in temperature and salinity vary among different model pairs, and increasing model resolution (from approximately 1° to about 0.1°) does not consistently lead to bias reduction in all regions for all models (Chassignet

et al., 2020). In a more recent study focusing on the simulated mixed layer depth (MLD) in these models, it was shown that increasing resolution can help reduce MLD biases in deep water formation regions, particularly in the Northern Hemisphere (Treguier et al., 2023). Neither of these high-resolution studies performed within the OMIP-2 framework specifically focused on the Arctic Ocean.

In this paper, we conducted an assessment of Arctic Ocean simulations using five pairs of matched low- and high-resolution

global ocean-sea ice models. These simulations were driven by the JRA55-do atmospheric state and runoff dataset (Tsujino et al., 2018) following the OMIP-2 protocol (Griffies et al., 2016). Unlike previous global model intercomparisons for Arctic Ocean simulations (Wang et al., 2016a, b; Ilicak et al., 2016; Shu et al., 2023), which focused on evaluating low-resolution models that are components of CMIP5 or CMIP6 models, the model pairs used in our study allowed us to specifically investigate the impact of model resolution. We evaluated the simulations concerning Arctic Ocean hydrography, the Atlantic Water layer,

stratification, freshwater content, and gateway transports.

The paper is structured as follows. In Section 2, we provide a brief description of the models used in this study. Section 3 is dedicated to evaluating the Arctic Ocean simulations and conducting comparisons between models and among model pairs. Finally, in Section 4, we summarize and discuss the results obtained from the evaluation.

## 2   Description of the model pairs

The models used in this study were forced by version 1.4.0 of the JRA55-do atmospheric forcing dataset (Tsujino et al., 2018) covering the period from 1958 to 2018. The OMIP protocol requires carrying out simulations with a long spin-up by repeating the forcing for at least 5 consecutive cycles (Griffies et al., 2016). However, due to the significant computational resources required for high-resolution simulations, previous high-resolution studies within the OMIP-2 framework, such as



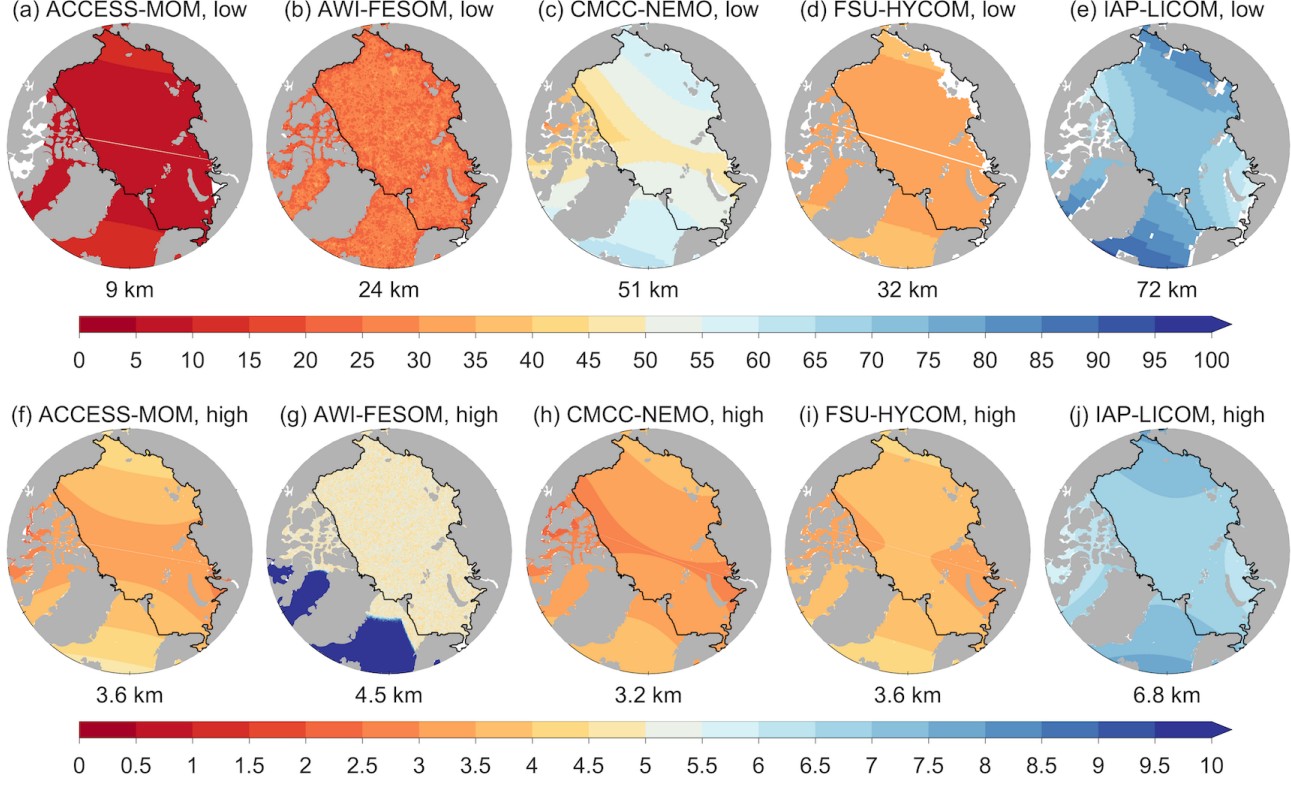

**Figure 2.** Model horizontal grid spacing (in kilometers) in five pairs of models: ACCESS-MOM, AWI-FESOM, CMCC-NEMO, FSU-HYCOM, and IAP-LICOM. The black contours indicate the area that is used to calculate the averaged grid size for the Arctic Ocean (denoted under each panel and shown in Table 1).

Chassignet et al. (2020) and Treguier et al. (2023), only considered the first cycle and acknowledged that the deep ocean

was still far from quasi-equilibrium. In line with these studies, we analyze the Arctic Ocean simulations in the first cycle of the OMIP-2 experiments, making it easier for model groups to participate. Model configurations, including resolutions and parameterizations, were determined by each model group based on their individual development practices. In this paper, the model results are based on monthly model output. Table 1 summarizes the five model pairs used in this study, and their corresponding horizontal resolutions are illustrated in Fig. 2.

ACCESS-MOM is the ocean and sea ice component of the Australian Community Climate and Earth System Simulator (ACCESS). It is based on MOM5.1 (Griffies, 2012) at 0.25° and 0.1° nominal horizontal grid spacing in the two configurations. These employ tripolar grids and the mean resolutions in the Arctic Ocean are 9 km and 3.6 km, respectively (Fig. 2). The vertical coordinate is $z^*$, with 50 and 75 levels, respectively. The configurations are described in detail in Kiss et al. (2020), with some updation described in the supplementary material of Solodoch et al. (2022). In both configurations, vertical mixing

is parameterized using the K-profile parameterization (KPP; Large et al., 1994) and a parameterization of submesoscale eddy effects in the surface mixed layer (FFH; Fox-Kemper et al., 2008, 2011) is employed. In addition, the Simmons et al. (2004)





**Table 1.** Model parameters for the low- and high-resolution configurations.

| Model | Horizontal grid | Vertical grid | Parameterizations in mixed layer | Sea surface salinity restoring |
|---|---|---|---|---|
| ACCESS-MOM low resolution | 1/4° tripolar (Arctic 9 km) | 50 z* levels top layer: 2.3 m | KPP, FFH | 33m per 300 d ($\Delta$s limited to 0.5 in flux calculation) |
| ACCESS-MOM high resolution | 1/10° tripolar (Arctic 3.6 km) | 75 z* levels top layer: 1.1 m | KPP, FFH | 33m per 300 d ($\Delta$s limited to 0.5 in flux calculation) |
| AWI-FESOM low resolution | 1° in most areas (Arctic 24 km) | 47 z-levels top layer: 5 m | KPP | 50m per 300 d (50m per 900 d in Arctic) |
| AWI-FESOM high resolution | 1° in most areas (Arctic 4.5 km) | 47 z-levels top layer: 5 m | KPP | 50m per 300 d (50m per 900 d in Arctic) |
| CMCC-NEMO low resolution | 1° tripolar (Arctic 51 km) | 50 z-levels top layer: 1 m | TKE | 100m per year (no restoring under ice) |
| CMCC-NEMO high resolution | 1/16° tripolar (Arctic 3.2 km) | 98 z-levels top layer: 1 m | TKE | 50m per year (no restoring under ice) |
| FSU-HYCOM low resolution | 0.72° tripolar (Arctic 32 km) | 41 hybrid layers | KPP | 30m per 60 d |
| FSU-HYCOM high resolution | 1/12° tripolar (Arctic 3.6 km) | 36 hybrid layers | KPP | 30m per 60 d |
| IAP-LICOM low resolution | 1° tripolar (Arctic 72 km) | 30 $\eta$ levels top layer: 10 m | Canuto scheme | 50m per 4 years (50m per 30 d under ice) |
| IAP-LICOM high resolution | 1/10° tripolar (Arctic 6.8 km) | 55 $\eta$ levels top layer: 5 m | Canuto scheme | 50m per 4 years (50m per 30 d under ice) |





bottom-enhanced internal tidal mixing and Lee et al. (2006) barotropic tidal mixing are included in both configurations. There is a spatially uniform background vertical diffusivity of $10^{-6}\,\mathrm{m^2 s^{-1}}$ at 0.1° but none at 0.25°. Redi (1982) diffusion and Gent and McWilliams (GM; Gent and McWilliams, 1990) parameterisation are used to represent the isoneutral diffusion and

thickness diffusivity due to unresolved eddies at 0.25°, but neither are used at 0.1°. The sea ice component of ACCESS-MOM is CICE5.1.2 (Hunke et al., 2015), with 5 thickness categories.

AWI-FESOM, the Finite element/volumE Sea ice-Ocean Model version 2 (Danilov et al., 2017), is a global unstructured-grid ocean general circulation model and forms the ocean and sea ice component of the Alfred Wegener Institute Climate Model (AWI-CM) (Sidorenko et al., 2019; Streffing et al., 2022). The model resolution is 1° in most global ocean areas and

refined to 24 km north of 45°N. The two configurations differ only in the horizontal resolution in the Arctic Ocean, with grid spacing of 24 km and 4.5 km, respectively. Both configurations employ 47 z-levels. Vertical mixing is parameterized using the KPP scheme, with background diffusivity of $4 \times 10^{-6}\,\mathrm{m^2 s^{-1}}$ in the Arctic region. Redi diffusion and the GM parameterization are employed, but deactivated in regions where the horizontal grid spacing is less than half the first baroclinic Rossby radius of deformation. The Redi diffusivity and GM coefficient are scaled with grid spacing in the horizontal and vary vertically

based on the squared buoyancy frequency (Ferreira et al., 2005; Danabasoglu and Marshall, 2007). The sea ice component of AWI-FESOM is FESIM2 (Danilov et al., 2015).

CMCC-NEMO, the Nucleus for European Modelling of the Ocean (NEMO) version 3.6 (Madec and the NEMO team, 2016), serves as the ocean and sea ice component of the CMCC climate model (CMCC-CM) (Cherchi et al., 2019). It employs tripolar grids with nominal horizontal resolutions of 1° and 1/16° for the two configurations. The corresponding mean resolutions are

51 km and 3.2 km in the Arctic Ocean. The model utilizes 50 and 98 z-levels in the two configurations, respectively. Vertical mixing coefficients are calculated using the Turbulent Kinetic Energy (TKE) parameterization introduced by Blanke and Delecluse (1993), which incorporates the effects of Langmuir cells and surface wave breaking (Madec and the NEMO team, 2016). The background vertical diffusivity is $1 \times 10^{-5}\,\mathrm{m^2 s^{-1}}$ and $1.2 \times 10^{-5}\,\mathrm{m^2 s^{-1}}$ in the low- and high-resolution configurations, respectively. In the low-resolution configuration, Redi and GM diffusivity coefficients are scaled with grid spacing, while

the high-resolution configuration employs biharmonic viscosity and diffusion for lateral mixing, with coefficients varying as the cube of the grid size (Iovino et al., 2023). The low-resolution configuration employed CICE4 (Hunke and Lipscomb, 2010) as its sea ice component, while the high-resolution configuration employed LIM2 (Timmermann et al., 2005).

FSU-HYCOM, a global version of the HYbrid Coordinate Ocean Model (HYCOM) (Chassignet et al., 2003), employs tripolar grids with horizontal resolutions of 0.72° and 1/12° for two configurations, corresponding to mean resolutions of

32 km and 3.6 km in the Arctic Ocean. The model employs 41 and 36 hybrid coordinate layers in the low and high horizontal resolution configurations, respectively. Vertical mixing is parameterized using the KPP scheme, with background diffusivity of $3 \times 10^{-5}\,\mathrm{m^2 s^{-1}}$. In the low-resolution configuration, interface height smoothing, equivalent to the GM diffusion, is achieved using a biharmonic operator with a mixing coefficient determined by the grid spacing multiplied by a velocity scale of $0.02\,\mathrm{m s^{-1}}$, except in the North Pacific and North Atlantic where a Laplacian operator with a velocity scale of $0.01\,\mathrm{m s^{-1}}$ is

employed. In the high-resolution configuration, interface height smoothing utilizes a biharmonic operator with a velocity scale of $0.015\,\mathrm{m s^{-1}}$. The sea ice component of FSU-HYCOM is CICE4 (Hunke and Lipscomb, 2010).




IAP-LICOM, the LASG/IAP Climate system Ocean Model (LICOM) version 3 (Li et al., 2020b; Lin et al., 2020), is the ocean and sea ice component of the Flexible Global Ocean-Atmosphere-Land System model (FGOALS) and the Chinese Academy of Sciences Earth System Model (CAS-ESM) (Li et al., 2020a; Bao et al., 2013). It employs tripolar grids with

nominal horizontal resolutions of approximately 1° and 1/10° for two configurations, resulting in mean resolutions of 72 km and 6.8 km in the Arctic Ocean. The model adopts the $\eta$ vertical coordinate (Mesinger and Janjic, 1985), utilizing 30 and 55 levels in the respective configurations. Mixing is parameterized using the scheme proposed by Canuto et al. (2002), with background diffusivity of $2 \times 10^{-6}\,\mathrm{m^2 s^{-1}}$. In addition, the St Laurent et al. (2002) tidal mixing scheme is employed. In the low-resolution configuration, isoneutral diffusion and GM parameterization are employed, with diffusivity coefficients scaled

vertically based on the squared buoyancy frequency (Ferreira et al., 2005). The sea ice component of IAP-LICOM is CICE4 (Hunke and Lipscomb, 2010). The high-resolution IAP-LICOM solely incorporates the thermodynamic part of CICE4, lacking its sea ice dynamics.

## 3   Results

### 3.1   Mean hydrography

#### 3.1.1   Temperature

We utilize the PHC3.0 hydrography climatology (Steele et al., 2001) to assess the basin-mean temperature and salinity. According to the PHC3.0 climatology, the warm Atlantic Water layer is situated beneath the cold surface water, spanning a depth range of approximately 150-900 m (Fig. 3). The warm core is in the depth range of 200-400 m and 400-600 m in the Eurasian and Amerasian basins, respectively. Since PHC3.0 primarily relies on observations from the 1970s to the 2000s, we compare

the model results averaged over the period from 1971 to 2000 to assess their agreement with PHC3.0.

In the Eurasian Basin, most of the low-resolution models, except for AWI-FESOM, underestimate the temperature within the core depth range of the Atlantic Water layer and overestimate the temperature below, extending to at least 2500 m depth (Fig. 3, upper panels). Conversely, three high-resolution configurations, namely ACCESS-MOM, CMCC-NEMO, and FSU-HYCOM, exhibit noticeable improvements with higher temperatures in the core depth range in the Eurasian Basin. The warm biases

in the deeper ocean are also reduced in ACCESS-MOM and CMCC-NEMO. Both configurations of AWI-FESOM faithfully represent the temperature within the core depth range of the Atlantic Water layer in the Eurasian Basin. The warm bias within the 500-1500 m depth range is lower in the high-resolution configuration of AWI-FESOM than its low-resolution configuration.

In the Amerasian Basin, the simulated maximum temperature aligns more closely with observations as the horizontal resolution increases, particularly in ACCESS-MOM, CMCC-NEMO, and FSU-HYCOM (Fig. 3, lower panels). However, in two of

these models (CMCC-NEMO and FSU-HYCOM), the high-resolution configuration exhibits larger warm biases below 500 m depth compared to the low-resolution configuration. In AWI-FESOM, with higher resolution, the warm bias below 500 m depth is reduced, although a slight cold bias emerges within the Atlantic Water core depth range.



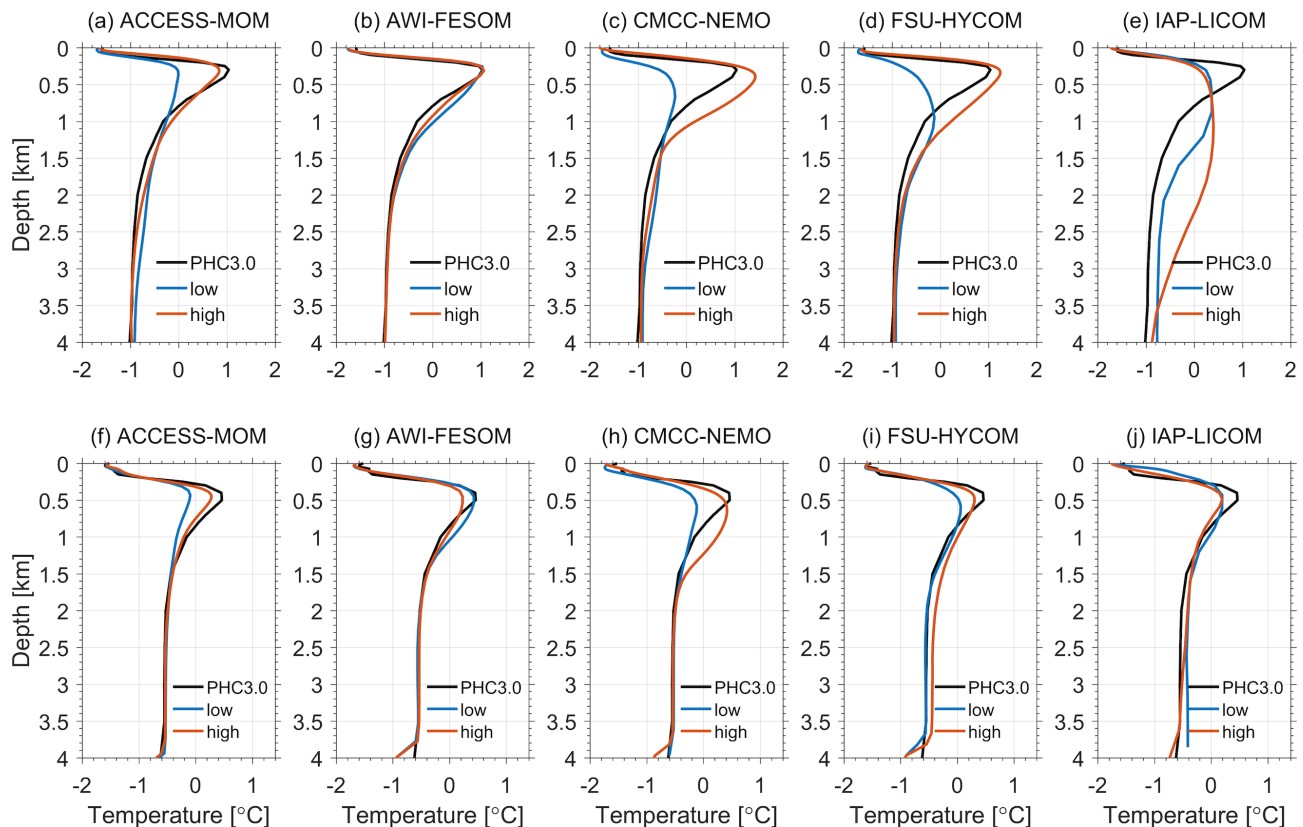

**Figure 3.** Basin-mean potential temperature profiles for the (a-e) Eurasian Basin and (f-j) Amerasian Basin from the five models at low (blue) and high (red) resolution, compared to the PHC3.0 hydrography climatology (black; Steele et al., 2001). The model results are averaged over 1971–2000. The Atlantic Water layer is characterized as the warm oceanic layer bounded by the 0°C isotherm.

The temperature maps at a depth of 400 m provide insight into the spatial distribution of the warm Atlantic Water in the deep basin of the Arctic (Fig. 4). Observational climatology shows that the warm Atlantic Water enters the Arctic basin through
Fram Strait and circulates in a cyclonic direction within the basin (Fig. 4k). However, most low-resolution models, except for AWI-FESOM, exhibit lower temperatures north of Svalbard compared to the observational climatology, indicating a deficiency in the inflow of warm Atlantic Water through Fram Strait in these models. Additionally, three of the low-resolution models (ACCESS-MOM, CMCC-NEMO, and FSU-HYCOM) show a prominent cold bias in the eastern Eurasian Basin (Fig. 4a,c,d). The maps of Atlantic Water core temperature (AWCT), representing the maximum temperature throughout the water column
in areas with bottom topography deeper than 150 m, demonstrate the absence of warm Atlantic Water in the eastern Eurasian Basin and its downstream region in these models (Fig. 5a,c,d). This cold bias can be traced back to the Barents Sea branch of the Atlantic Water inflow, where the temperature is much colder in these three models compared to other models and their high-resolution counterparts. Hence, the cold biases in the deep basin of the Arctic can be attributed to both insufficient inflow of warm water through Fram Strait and excessive discharge of cold water from the St. Anna Trough, consistent with findings





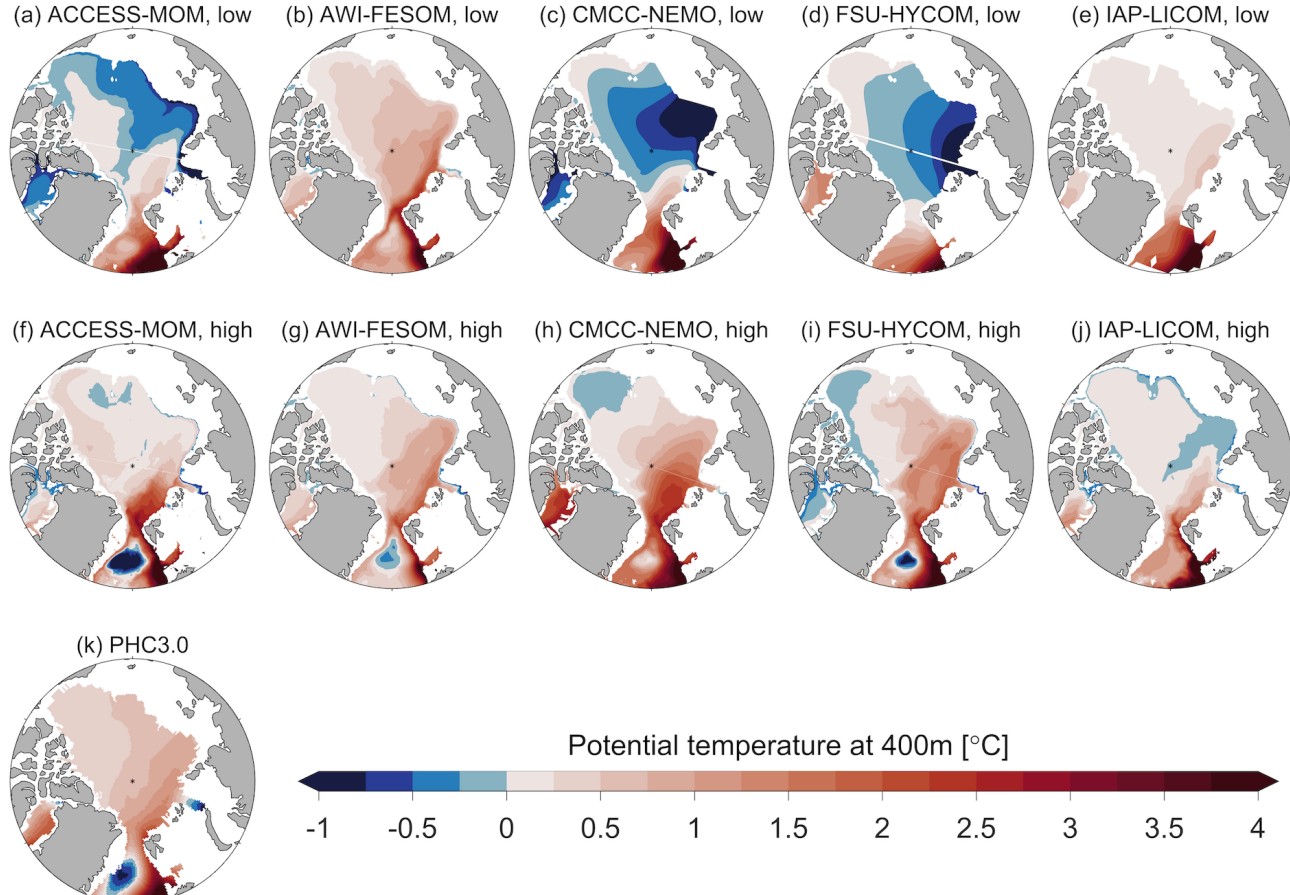

**Figure 4.** Spatial distribution of the simulated potential temperature at 400 m depth averaged over 1971–2000: (a-e) low-resolution models versus (f-j) high-resolution models. The PHC3.0 temperature climatology (Steele et al., 2001) at 400 m depth is shown in (k).

from previous model intercomparison studies (Ilicak et al., 2016; Shu et al., 2019). In the high-resolution configurations, both issues are mitigated, resulting in a significant reduction of the cold bias in the deep basin (Fig. 5f,h,i).

Both the temperature at 400 m depth and the AWCT demonstrate that AWI-FESOM reasonably represents the Atlantic Water boundary current along the Eurasian continental slope in both configurations (Figs. 4b,g and 5b,g). In its high-resolution configuration, the Atlantic Water boundary current extends all the way to the Laptev Sea, with a portion of it recirculating

along the Lomonosov Ridge (Fig. 5g), which is more consistent with observations (Woodgate et al., 2001). FSU-HYCOM's high-resolution configuration also captures the Atlantic Water boundary current (Fig. 5i). However, the high-resolution configurations of ACCESS-MOM and CMCC-NEMO exhibit a broad Atlantic Water flow from Fram Strait into the Eurasian Basin instead of a distinct boundary current along the continental slope (Figs. 4f,h and 5f,h). In the case of IAP-LICOM, increasing the resolution results in a warmer Atlantic Water layer north of Svalbard, but it remains colder than observed, indicating an

insufficient inflow through Fram Strait (Figs. 4g and 5g).





**Figure 5.** Simulated Atlantic Water core temperature (AWCT) averaged over 1981–1995: (a-e) low-resolution versus (f-j) high-resolution simulations. (k) The AWCT for the same period based on observations (Polyakov et al., 2020).

Fig. 6 depicts the vertical transect of temperature across the Arctic Ocean. According to the PHC3.0 climatology, the warm Atlantic Water layer exhibits a deepening upper boundary ($0°C$ isotherm) from the Eurasian to the Amerasian Basin (Fig. 6k). It also indicates that the intermediate and deep layers are warmer in the Amerasian Basin compared to the Eurasian Basin. The latter is because cold dense water originating from the Barents Sea penetrates into the Eurasian Basin, but cannot

pass the Lomonosov Ridge to enter the Amerasian Basin. Among the low-resolution models, only AWI-FESOM successfully simulates a warm Atlantic Water layer with a depth range and temperature magnitude similar to the observation (Fig. 6b). In the low-resolution configuration of IAP-LICOM, an Atlantic Water layer extends from the Eurasian to the Amerasian Basin, but its shape and temperature magnitude differ from the observation (Fig. 6e). Encouragingly, ACCESS-MOM, CMCC-NEMO, and FSU-HYCOM demonstrate the ability to simulate the Atlantic Water layer more accurately when their resolutions

are increased, despite some biases in thickness (i.e., too thin in ACCESS-MOM and too thick in CMCC-NEMO and FSU-HYCOM) (Fig. 6f,h,i). The temperature transects in the two configurations of AWI-FESOM are comparable, with the upper



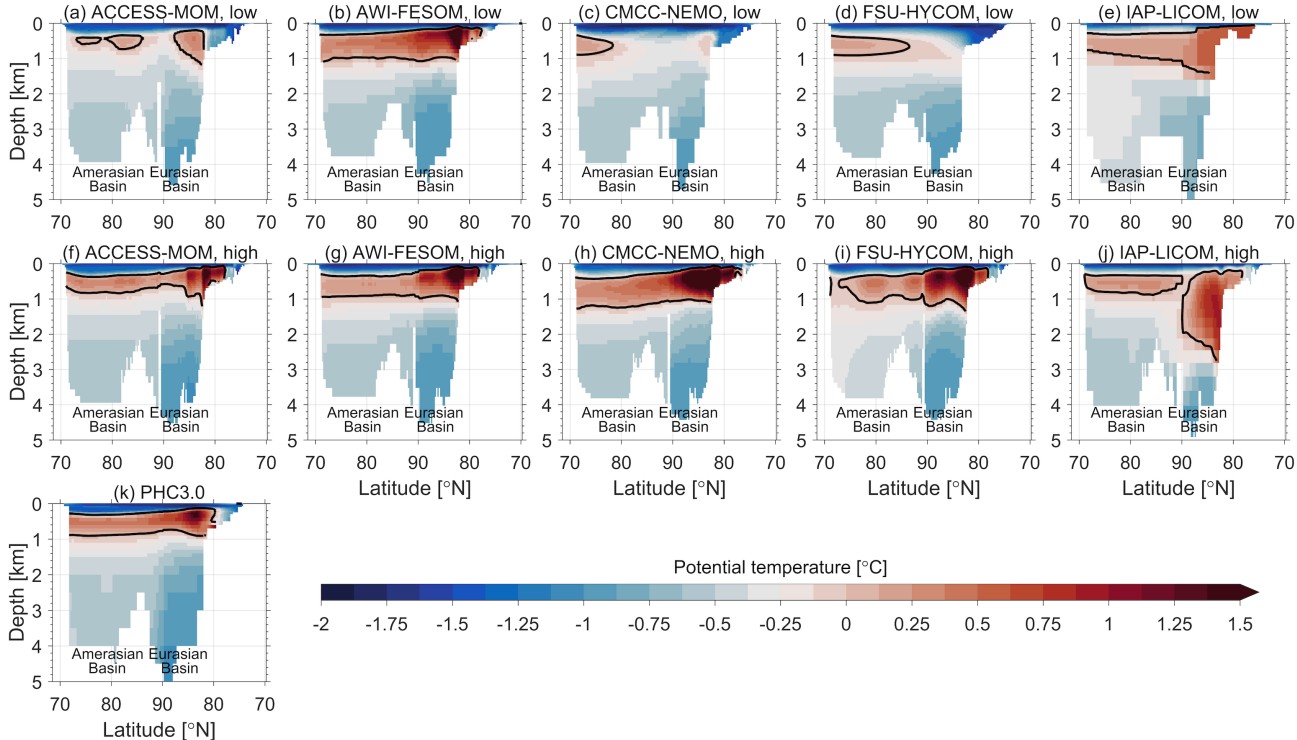

**Figure 6.** Potential temperature in a vertical transect crossing the Arctic basin averaged over 1971–2000: (a-e) low-resolution versus (f-j) high-resolution simulations. The PHC3.0 temperature climatology (Steele et al., 2001) is shown in (k). The boundary of the Atlantic Water layer, the 0°C isotherm, is indicated by black contour lines. The transect is along the longitudes of 145°W and 70°E, and its location is indicated in Fig. 1.

boundary depth and thickness of the Atlantic Water layer being more realistic in the high-resolution configuration, although the AWCT in the Amerasian Basin is slightly biased low in this case (Fig. 6b,g).

All the models successfully reproduce the temperature contrast in the intermediate and deep layers between the two basins
(Fig. 6). However, the high-resolution IAP-LICOM exhibits an excessively thick Atlantic Water layer in the Eurasian Basin (Fig. 6j). Additionally, its Atlantic Water layer is split into two cells due to a cold tongue recirculating along the Lomonosov Ridge (Fig. 4j). The degradation of the IAP-LICOM simulation at high resolution is likely due to the misrepresentation of sea ice resulting from the absence of sea ice dynamics in the model (Chassignet et al., 2020). Below 1000 m depth in the Amerasian Basin, the high-resolution FSU-HYCOM exhibits a notable warm bias that is absent in its low-resolution counterpart (Fig. 6i).
This bias might be due to the lower vertical resolution in the high-resolution configuration of FSU-HYCOM than in its low-resolution configuration (Table 1).



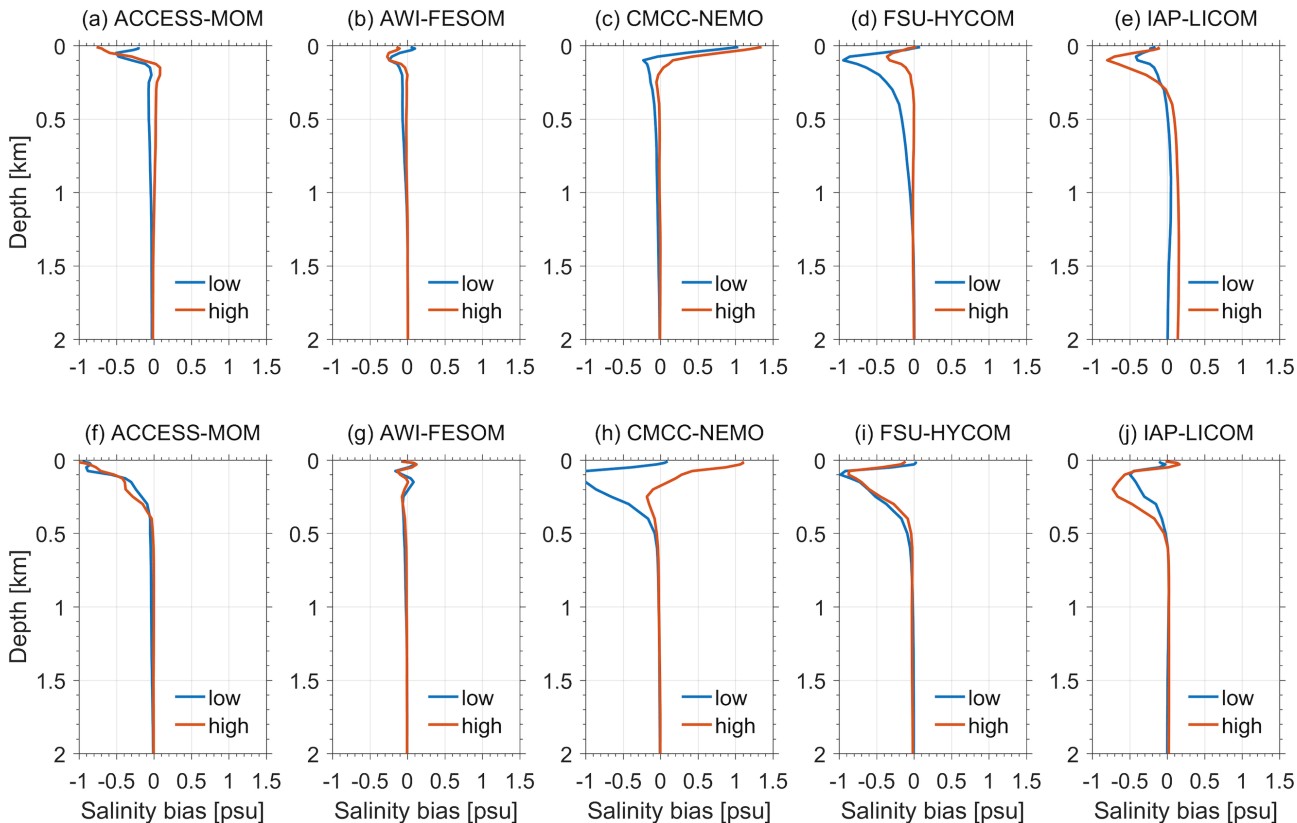

**Figure 7.** Biases of basin-mean salinity for (a-e) the Eurasian Basin and (f-j) the Amerasian Basin in five models at low (blue) and high (red) resolution. The results are averaged over 1971–2000 and relative to the PHC3.0 climatology (Steele et al., 2001). The profiles of basin-mean salinity in the models and PHC3.0 are shown in Figure S1.

### 3.1.2 Salinity

Fig. 7 illustrates the model biases in salinity profiles. All the models tend to exhibit a negative salinity bias in the halocline below the surface layer. This bias is likely caused by excessive vertical mixing in the models, which reduces salinity in the halocline and increases it near the surface (Wang et al., 2018). To mitigate the issue of large drift in ocean salinity and circulation, global models typically restore sea surface salinity to climatology (Griffies et al., 2009). The restoring can dampen the increase in surface salinity induced by vertical mixing. As a result of sea surface salinity restoring and strong vertical mixing, the mean salinity is underestimated, as evident from the overestimation of liquid freshwater content (see Section 3.4). This issue was previously investigated in the CORE-II Arctic Ocean study (Wang et al., 2016a), and it appears that the state-of-the-art ocean models in OMIP-2 still encounter the same challenge as the CORE-II models.

It is encouraging to observe that the high-resolution configurations exhibit smaller salinity biases in the halocline in all models except for IAP-LICOM, particularly in the Eurasian Basin (Fig. 7). Previous studies have suggested that an inadequate



**Figure 8.** Simulated salinity at 400 m depth averaged over 1971–2000 (a-j). The PHC3.0 salinity climatology (Steele et al., 2001) at 400 m depth is shown in (k).

treatment of brine rejection could lead to static instability and excessive vertical mixing over a wide depth range, resulting in a negative salinity anomaly in the halocline and a positive salinity anomaly at the surface (Nguyen et al., 2009). However,

our findings indicate that increasing model resolution can reduce the negative salinity bias in the halocline, suggesting that at least part of this bias is unrelated to the treatment of brine rejection in the models, as none of the models analyzed in this study employ brine rejection parameterization for the Arctic Ocean. CMCC-NEMO and ACCESS-MOM exhibit relatively large positive and negative salinity biases at the ocean surface, respectively, and these biases are even amplified in their high-resolution configurations (Fig. 7a,c,f,h), which could be attributed to the limited sea surface salinity restoring in these models

(Table 1). IAP-LICOM displays larger positive or negative salinity biases throughout the ocean column in its high-resolution configuration compared to the low-resolution configuration (Fig. 7e,j). In both basins, AWI-FESOM exhibits the smallest salinity bias among the models, and its bias is similar between the two configurations (Fig. 7b,g).





Similar to the distribution of temperature (Fig. 4k), the distribution of salinity at 400 m depth illustrates the cyclonic circulation of the Atlantic Water along the continental slope (Fig. 8k). The Canada Basin displays the lowest salinity at this depth,

reflecting the deepening of the isohaline due to Ekman convergence induced by the Beaufort High sea level pressure. Most of the model simulations are able to capture the basic salinity contrast between the Eurasian Basin and Amerasian Basin (Fig. 8). The low-resolution configuration of FSU-HYCOM exhibits a significant negative bias in salinity throughout the deep basin at 400 m depth (Fig. 8d). It fails to simulate the Atlantic Water boundary current entering the basin through the Fram Strait, which carries warm, saline Atlantic Water in reality. However, its high-resolution configuration shows an improved representation of

the Atlantic Water inflow and, consequently, a better representation of salinity in the Eurasian Basin (Fig. 8i). Nevertheless, its salinity in the Canada Basin remains biased low. The high-resolution configurations of ACCESS-MOM and CMCC-NEMO also demonstrate better simulation of salinity in the Eurasian Basin compared to their low-resolution counterparts, but their salinity in the Canada Basin is still biased low (Fig. 8f,h), similar to the high-resolution FSU-HYCOM. In AWI-FESOM, the cyclonic circulation of the Atlantic Water is better simulated with higher resolution (Fig. 8b,g). Some of the Atlantic Water

directly penetrates towards the North Pole and Amerasian Basin in its low-resolution configuration, and this issue is resolved in the high-resolution configuration. As the vertical resolution is the same in both AWI-FESOM configurations, the improved model performance can be attributed to higher horizontal resolution. The salinity bias in IAP-LICOM is more pronounced in its high-resolution configuration for both basins (Fig. 8e,j), likely due to the impact of misrepresented sea ice cover.

Despite the improvements in representing salinity in high-resolution models as described above, it is important to acknowl-
edge that the salinity biases in most of these models (particularly in the halocline and/or surface layer) still exceed the magnitudes of changes observed over decades, as shown in Fig. 9. In several models with significant salinity biases (up to approximately 1 psu), these biases quickly escalate to high levels within the initial few years of the model simulations. In certain cases, such as the low-resolution FSU-HYCOM model, the fresh biases persist and extend downwards throughout the entire simulation period (Fig. 9d,n). Background vertical diffusivity employed in models can significantly influence the vertical distribution

of salinity and the stratification in the Arctic Ocean (Zhang and Steele, 2007). The underlying cause for the larger fresh biases in the halocline of FSU-HYCOM compared to AWI-FESOM could partially be attributed to the background diffusivity within the KPP mixing scheme. In FSU-HYCOM, the background diffusivity is $3 \times 10^{-5}\,\mathrm{m^2 s^{-1}}$, which is approximately one order of magnitude higher than that of AWI-FESOM ($4 \times 10^{-6}\,\mathrm{m^2 s^{-1}}$). However, in the case of IAP-LICOM, which has a relatively small background diffusivity of $2 \times 10^{-6}\,\mathrm{m^2 s^{-1}}$, the fresh biases remain substantial (Fig. 9e,j,o,t). Therefore, it is evident that

other factors, such as explicit mixing from applied parameterizations and numerical mixing, also contribute to the salinity biases.

## 3.2 Warming events in Atlantic Water layer

Observations have revealed several warming events in the Arctic Atlantic Water layer, which are associated with strengthened ocean heat influx through the Fram Strait. These events occurred in the 1990s and the 2010s (Steele and Boyd, 1998; Gerdes

et al., 2003; Karcher et al., 2012; Polyakov et al., 2012, 2020; Wang et al., 2020b). The abnormally high North Atlantic Oscillation in the 1990s strengthened the Atlantic Water boundary current in the Nordic Seas and increased the inflow through



**Figure 9.** Depth-time plot of basin-mean salinity deviation from PHC3.0 climatology in the (upper two rows) Eurasian Basin and (lower two rows) Amerasian Basin. (a)-(e) and (k)-(o) are for the low-resolution models. (f)-(j) and (p)-(t) are for the high-resolution models.

the Fram Strait (Dickson et al., 2000). Simultaneously, the positive Arctic Oscillation strengthened the cyclonic circulation within the Arctic Ocean and facilitated the influx of Atlantic Water from the Fram Strait (Wang et al., 2023). During the 2010s, both the warming in the inflow water and the intensified inflow through the Fram Strait due to Arctic sea ice decline contributed to the significant warming of the Atlantic Water layer in the Arctic basin (Wang et al., 2020b). It is crucial to assess whether the OMIP-2 models, driven by the same atmospheric forcing, are capable of reasonably reproducing these events.

Fig. 10 presents the depth-time plot of basin-mean temperature in the Eurasian and Amerasian basins. In the low-resolution models, AWI-FESOM successfully reproduces the warming events in the Eurasian Basin (Fig. 10b). ACCESS-MOM and CMCC-NEMO exhibit signals of these warming events in their low-resolution configurations, but with lower magnitudes (Fig. 10a,c). This is consistent with their cold bias in simulated mean temperature (Figs. 4 and 5). The 1990s warming is absent in the low-resolution FSU-HYCOM and IAP-LICOM (Fig. 10d,e).



**Figure 10.** Depth-time plot of basin-mean potential temperature of the (upper two rows) Eurasian Basin and (lower two rows) Amerasian Basin. (a)-(e) and (k)-(o) are for the low-resolution models. (f)-(j) and (p)-(t) are for the high-resolution models.





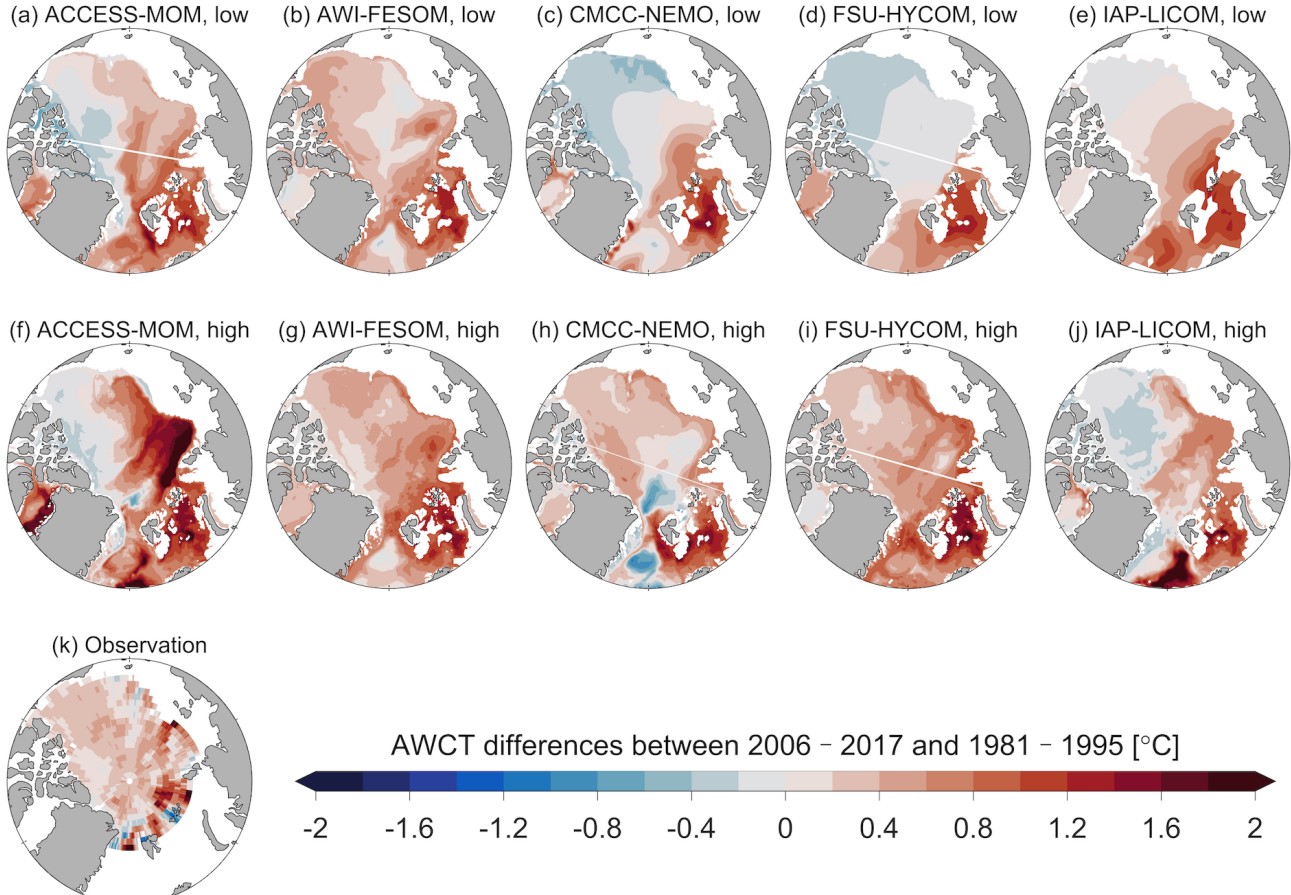

**Figure 11.** Difference of the Atlantic Water core temperature (AWCT) between 2006–2017 and 1981–1995 in the models (a-j) and observations (k) (Polyakov et al., 2020). The AWCT in these two periods is shown in Fig. 5 and Figure S2, respectively.

Among the high-resolution models, with the exception of IAP-LICOM, all are capable of reproducing the two warming events in the Eurasian Basin (Fig. 10f-j). However, there is a notable difference between AWI-FESOM and other models: the deepening trend of the lower boundary of the warm Atlantic Water layer is smaller in AWI-FESOM, indicating a smaller model drift at the intermediate depth (Fig. 10f-j). Comparing the two configurations of AWI-FESOM reveals that increasing horizontal resolution leads to better simulation of the thickness of the Atlantic Water layer and reduced warming drift at the intermediate depth (Fig. 10b,g), consistent with previous findings (Wang et al., 2018).

The warming in the Eurasian Basin propagates into the Amerasian Basin with a time lag of a few years (Steele and Boyd, 1998; Polyakov et al., 2012). Since most of the low-resolution models fail to accurately reproduce the two warming events in the Eurasian Basin, they do not exhibit both warming events in the Amerasian Basin (Fig. 10k,m-o). In contrast, all the high-resolution configurations, except for IAP-LICOM, can simulate the warming events in the Amerasian Basin, with a time lag of about 4 years compared to the Eurasian Basin (Fig. 10p-s).





Hydrography observations in the Arctic Ocean are relatively sparse in time and space, leading to significant uncertainty in the gridded temperature data based on these observations. With this limitation in mind, we utilize the gridded AWCT averaged over two periods (1981–1995 and 2006–2017) as described in Polyakov et al. (2020) to evaluate the simulated AWCT changes in the models. Fig. 11 presents the difference in AWCT between these two periods for both the observation and the model simulations. The observations indicate a clear increase in AWCT in most areas of the Arctic basin (Fig. 11k). However, four out of the five low-resolution models simulate a reduction in AWCT, covering large spatial extents. FSU-HYCOM exhibits a decrease in AWCT across nearly the entire Arctic basin (Fig. 11d), while CMCC-NEMO shows a reduction in most of the Amerasian Basin (Fig. 11c), and ACCESS-MOM and IAP-LICOM display a decrease in part of the Amerasian Basin (Fig. 11a,e).

The high-resolution FSU-HYCOM demonstrates an evident improvement in simulating the AWCT change (Fig. 11i). Although the high-resolution CMCC-NEMO better represents the AWCT change in the Amerasian Basin compared to its low-resolution counterpart, it exhibits an erroneous cooling anomaly in the Eurasian Basin (Fig. 11h), potentially attributed to the excessively strong warming in the 1990s simulated by high-resolution CMCC-NEMO (Fig. 10h). Neither ACCESS-MOM nor IAP-LICOM show significant improvement in simulating the rise of AWCT in the Amerasian Basin in their high-resolution configurations (Fig. 11f,j). These models seem to struggle with advecting the signal of Atlantic Water warming into the Amerasian Basin, which could be explained by the presence of a too large and strong anticyclonic Beaufort Gyre indicated by the excess freshwater content (see Section 3.4). The upper ocean circulation has a strong imprint on the Atlantic Water layer circulation (Lique et al., 2015; Hinrichs et al., 2021; Wang et al., 2023).

### 3.3 Mixed layer depth and cold halocline base depth

The winter mixed layer depth (MLD) in the Barents Sea is deeper than in the Arctic deep basin (Peralta-Ferriz and Woodgate, 2015), reflecting the strong heat loss from the warm Atlantic Water to the atmosphere in the Barents Sea (Schauer et al., 1997; Smedsrud et al., 2013; Shu et al., 2021). In the Arctic deep basin, the MLD remains relatively shallow even during winter due to the presence of low salinity water at the surface. The winter MLD is not only a climate-relevant variable but also an important factor that regulates summer primary production in the Arctic (Popova et al., 2010). Between the surface mixed layer and the Atlantic Water layer lies the Arctic halocline, which acts as an insulating layer, inhibiting the transfer of heat from the Atlantic Water layer to the cold mixed layer and sea ice. An uplift of the boundary between the halocline and Atlantic Water layer, accompanied by a weakening of the halocline stratification and warming of the Atlantic Water layer, has been observed in the eastern Eurasian Basin in the 2010s (Polyakov et al., 2017, 2020). This phenomenon, known as Arctic Atlantification (Polyakov et al., 2017), is primarily driven by the decline in Arctic sea ice (Wang et al., 2020b). In the following we will evaluate the simulations of the MLD and halocline base depth in the models.

### 3.3.1 Winter MLD

The MLD is defined as the depth at which the potential density exceeds the surface density by $0.1\,\mathrm{kg\,m^{-3}}$, a threshold found to be most suitable for the Arctic Ocean MLD (Peralta-Ferriz and Woodgate, 2015). We compute the MLD using monthly-



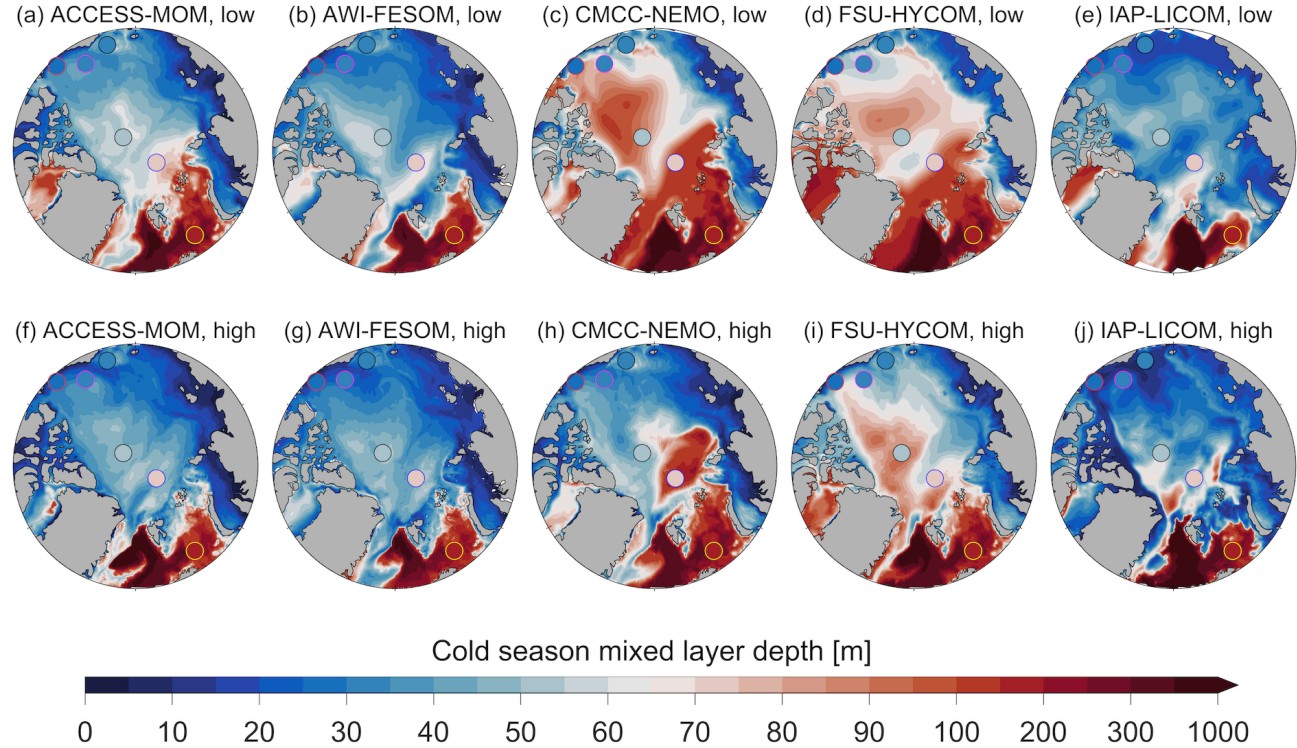

**Figure 12.** Mixed layer depth (MLD) in the cold season (November to May) averaged over 1979–2012. The observational estimates for six regions are shown as filled circles (Peralta-Ferriz and Woodgate, 2015). The color scale is nonlinear.

mean temperature and salinity from the models, while noting the cautionary remarks in Treguier et al. (2023) about how MLD calculated from monthly-mean data will differ from higher-frequency data. Fig. 12 depicts the MLD averaged over November to May during the period 1979–2012 for each model. Observational estimates of the winter MLD in six Arctic regions, based on hydrography observations (Peralta-Ferriz and Woodgate, 2015), are also shown as circles. The observational estimates indicate

that the winter MLD is approximately 30 m in the southern Beaufort Sea, Canada Basin, and Chukchi Sea, about 50 m in the Makarov Basin, around 70 m in the Eurasian Basin, and roughly 170 m in the Barents Sea.

ACCESS-MOM, AWI-FESOM, and IAP-LICOM can reproduce the contrast between the deep MLD in the Barents Sea and the shallow MLD in the Arctic deep basin (Fig. 12a,b,e). Increasing horizontal resolution leads to a reduction in MLD in the Arctic deep basin in these models except for the Eurasian Basin of IAP-LICOM (Fig. 12f,g,j). Mesoscale eddies have an

effect on restratifying the mixed layer, thereby reducing the MLD (Treguier et al., 2023). The resolutions used in the high-resolution OMIP-2 configurations (3-6 km, Fig. 2) are only eddy-permitting in the Arctic deep basin (Wang et al., 2020a). The comparison in Fig. 12 indicates that the high-resolution configurations can capture some of the eddy effects, although eddies are not fully resolved yet. The low-resolution ACCESS-MOM and AWI-FESOM models provide a good representation of the winter MLD in the Arctic deep basin. In their high-resolution counterparts, the MLD is approximately 10-20 m shallower and





slightly underestimates the observations. However, as the MLD computed from monthly temperature and salinity tends to be shallower than that computed from snapshot profiles due to the nonlinearity of the MLD (Treguier et al., 2023), we should not conclude that these high-resolution configurations have worse MLD than the low-resolution ones.

The low-resolution CMCC-NEMO and FSU-HYCOM models simulate too deep MLD in both the Eurasian and Amerasian basins (Fig. 12c,d). This overestimation can be attributed to stratification biases in the upper ocean within these models. Specif-

ically, they demonstrate either positive salinity biases at the surface (see Fig. 9c) or negative salinity biases in the subsurface (see Fig. 9d,m,n). Such salinity biases lead to reduced stratification, consequently promoting the formation of deep mixed layers during wintertime. Our finding is consistent with previous research, which highlighted the dominating impact of the simulated salinity profile, and consequently density stratification, on models' performance in simulating winter MLD (Allende et al., 2023). In the high-resolution CMCC-NEMO and FSU-HYCOM models, there is a partial improvement in the MLD

estimation within certain regions of the Arctic deep basin (Fig. 12h,i). However, this improvement does not correspond to a reduction in salinity biases. For example, the significant fresh bias observed in the subsurface of the Amerasian Basin in the low-resolution CMCC-NEMO model is replaced by a positive salinity bias at the surface in its high-resolution counterpart (Fig. 9m,r). The salinity biases in different depth ranges altered in such a manner that the overall upper ocean stratification in the Amerasian Basin is enhanced, resulting in shallower MLD than in the low-resolution configuration (Fig. 12h). Similarly, the

decrease in the MLD in the high-resolution FSU-HYCOM model (Fig. 12i) can be partially explained by the amplified fresh bias at surface (comparing Fig. 9i,s with Fig. 9d,n).

Additionally, we computed the MLD in March using the density threshold of $0.03\,\mathrm{kg m^{-3}}$ and made the comparison with the MIMOC MLD dataset (Schmidtko et al., 2013), which also used this threshold. This comparison yields similar findings to those described above (Figure S3).

**3.3.2 Cold halocline base depth**

The cold halocline base depth is defined as the depth of the $0\,°\mathrm{C}$ isotherm between the halocline and Atlantic Water layer (Polyakov et al., 2020). It deepens from the Eurasian Basin toward the Canada Basin (Fig. 13k). In the low-resolution ACCESS-MOM, CMCC-NEMO, and FSU-HYCOM models, where there is no Atlantic Water warmer than $0\,°\mathrm{C}$ in a large area of the Arctic deep basin (Fig. 5 and Figure S2), the cold halocline base depth cannot be defined (Fig. 13a,c,d). With the improved

representation of ocean temperature in the high-resolution configurations of these models, the cold halocline base depths show a spatial pattern similar to the observations, although there is a deep bias in the Amerasian Basin (Fig. 13f,h,i). Both configurations of the AWI-FESOM model reasonably reproduce the spatial pattern and magnitudes of the cold halocline base depth (Fig. 13b,g).

Observations have shown a shoaling of the cold halocline base depth in most of the Arctic deep basin during the period

2006–2017 compared to 1981–1995 (Fig. 14k, Polyakov et al., 2020). However, the three models that showed improvement in simulating the mean state of the cold halocline base depth with higher resolution (ACCESS-MOM, CMCC-NEMO, and FSU-HYCOM) do not reproduce the observed shoaling in the Eurasian Basin or Canada Basin (Fig. 14f,h,i). Both configurations of the AWI-FESOM model simulate an uplift of the cold halocline base depth in the Eurasian Basin, with magnitudes very





**Figure 13.** Cold halocline base depth averaged over 2006–2017 in low-resolution (a)-(e) and high-resolution (f)-(j) models. The observational estimate is shown in (k) (Polyakov et al., 2020).

similar to the observations (Fig. 14b,g). However, its high-resolution configuration exhibits a large overestimation of the uplift
in the Canada Basin (Fig. 14g). The overestimation of the uplift in the Canada Basin is mainly due to the deep bias in the cold
halocline base depth in the earlier period (1981–1995, Figure S4) since the model reproduces the cold halocline base depth well
in the recent period (2006–2017, Fig. 13g). In the high-resolution IAP-LICOM model, the strong deepening of the Atlantic
Water layer over time in the Amerasian Basin (Fig. 10t) leads to a downwelling of the cold halocline base depth in that region
(Fig. 14j).

All the high-resolution models that simulate the warming of the Atlantic Water layer in the 2010s show an uplift of the
cold halocline base depth in the Eurasian Basin during that period (Fig. 10f-i). Thus, these models are able to reproduce the
fact that the warm Atlantic Water layer has become closer to the surface in the progression of Arctic Atlantification in the
2010s. However, the cold halocline base depth simulated in the high-resolution CMCC-NEMO and FSU-HYCOM models is
too shallow in the 1990s, associated with the warming event during that period (Fig. 10h,i). This causes the cold halocline





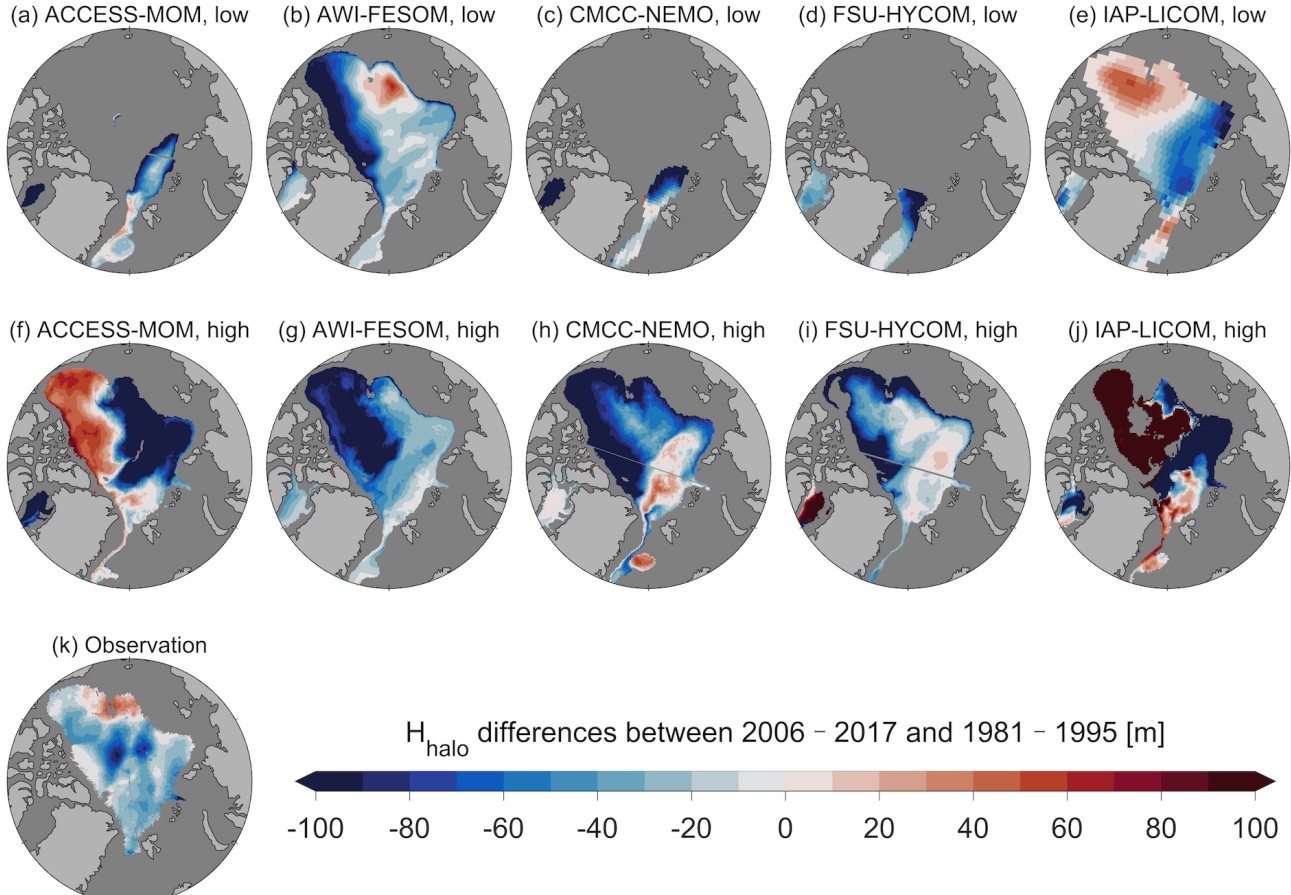

**Figure 14.** Change of the cold halocline base depth between the period 2006–2017 and the period 1981–1995 in low-resolution (a)-(e) and high-resolution (f)-(j) models. The observational estimate (Polyakov et al., 2020) is shown in (k).

base depth in the Eurasian Basin in 2006–2017 to be slightly deeper than in 1981–1995 in these two models (Fig. 14h,i), contradicting the observations.

### 3.4 Liquid freshwater content

The Arctic Ocean plays a crucial role in the hydrological cycle of the Northern Hemisphere (Carmack et al., 2016). It receives freshwater from various sources, including river runoff, net precipitation, and low salinity Pacific Water, while exporting fresh-
water to the subpolar North Atlantic. The Beaufort High, characterized by high sea level pressure, causes the freshwater in the Arctic to accumulate predominantly in the Canada Basin (McPhee et al., 2009; Proshutinsky et al., 2009, 2019; Timmermans and Marshall, 2020; Wang and Danilov, 2022). Due to the prevailing anticyclonic wind patterns and the decline of Arctic sea ice, the Arctic Ocean has been experiencing an increase in liquid freshwater content since the mid-1990s (Proshutinsky et al., 2019; Wang and Danilov, 2022). Observations have revealed that the amount of liquid freshwater in the Arctic basin in the





mid-2010s was approximately $11,000\,\mathrm{km}^3$ more than in the mid-1990s (Rabe et al., 2014; Wang et al., 2019). The excess freshwater in the Arctic, when released into the convective regions of the North Atlantic, could impact deep water formation and large-scale circulation (Aagaard et al., 1985; Goosse et al., 1997; Arzel et al., 2008). Therefore, assessing the Arctic freshwater content is important for understanding climate dynamics.

The freshwater content of the water column, referred to as the freshwater column in short (measured in meters), is defined as follows:

$$\mathrm{FWC} = \int_H^0 (S_{ref} - S)/S_{ref}\,\mathrm{d}z, \tag{1}$$

where $S$ represents salinity, $S_{ref}$ is the reference salinity, and $H$ is the depth at which the salinity equals the reference salinity. It quantifies the amount of pure water that needs to be removed from a column to change the mean salinity to the reference salinity. In this study, a reference salinity of $S_{ref} = 34.8\,\mathrm{psu}$, considered the mean salinity of the Arctic Ocean (Aagaard and Carmack, 1989), is used, consistent with previous studies (e.g., Serreze et al., 2006; Jahn et al., 2012; Haine et al., 2015; Wang et al., 2016a, 2023; Shu et al., 2023). The volumetric freshwater content is obtained by integrating the freshwater column over an area.

First, we evaluate the mean state of the simulated freshwater column (Fig. 15). The models generally capture the basic spatial pattern of the freshwater column, with higher values in the Canada Basin and lower values in the Eurasian Basin. However, there are notable differences in the spatial distribution and magnitudes of the freshwater column among the models. In the low-resolution models, CMCC-NEMO and FSU-HYCOM tend to significantly overestimate the freshwater column in the Amerasian Basin (Fig. 15c,d), while AWI-FESOM underestimates the freshwater column in the northwestern Amerasian Basin (Fig. 15b).

The high-resolution configurations do not exhibit a significant improvement in simulating the mean state of the freshwater column. In the high-resolution models, ACCESS-MOM shows an even stronger overestimation of the freshwater column in the Amerasian Basin compared to its low-resolution counterpart (Fig. 15a,f). AWI-FESOM remains largely similar between the two configurations (Fig. 15b,g), while CMCC-NEMO underestimates the freshwater column in the high-resolution configuration, contrary to its overestimation in the low-resolution configuration (Fig. 15c,h). FSU-HYCOM displays an excessive concentration of freshwater in the southern Beaufort Sea in its high-resolution configuration (Fig. 15i), and IAP-LICOM fails to reproduce a realistic gyre shape in the Amerasian Basin's freshwater distribution (Fig. 15j). As the freshwater column plays a crucial role in determining the sea surface height and the surface geostrophic current in the Arctic basin (Armitage et al., 2017; Wang, 2021), these results also indicate a large spread in the simulated mean state of ocean surface circulation among the models.

Next, we will assess the model spread in the Arctic basin freshwater content. Fig. 16 presents the time series of freshwater content in the Arctic basin and their anomalies relative to the 1992–2008 mean. Consistent with the mean state of the freshwater column shown in Fig. 15, the model spread in simulating Arctic basin freshwater content remains similar in the high-resolution configurations to the low-resolution configurations (Fig. 16a,c). In the low-resolution CMCC-NEMO and FSU-HYCOM models, the freshwater content in the Arctic basin drifts upward over time (Fig. 16a). The most significant drift occurs during



(a) ACCESS-MOM, low  (b) AWI-FESOM, low  (c) CMCC-NEMO, low  (d) FSU-HYCOM, low  (e) IAP-LICOM, low

(f) ACCESS-MOM, high  (g) AWI-FESOM, high  (h) CMCC-NEMO, high  (i) FSU-HYCOM, high  (j) IAP-LICOM, high

(k) PHC3.0

Liquid freshwater column [m]

0  2  4  6  8  10  12  14  16  18  20  22  24

**Figure 15.** Liquid freshwater column (in meters) averaged over 1971–2000 in (a-e) low-resolution and (f-j) high-resolution models. The estimate based on PHC3.0 (Steele et al., 2001) is shown in (k).

the first 10 years of the simulation, as also indicated in the time-depth plot of salinity (Fig. 9). In the high-resolution FSU-

415 HYCOM model, the upward drift of total freshwater content is reduced (Fig. 16a,c), mainly attributed to the lower freshwater column outside the Beaufort Sea (Fig. 15d,i). The high-resolution CMCC-NEMO model simulates a downward drift in freshwater content during the first 40 years (Fig. 16c), which is associated with the evolution of positive salinity bias in the upper Amerasian Basin in terms of both magnitude and vertical extent (Fig. 9r). The high-resolution IAP-LICOM model, unlike its low-resolution counterpart, exhibits a strong upward drift (Fig. 16a,c).

Lastly, we will assess the simulation of temporal changes in the Arctic freshwater content. Except for IAP-LCOM, all models consistently simulate an increase in Arctic basin freshwater content during the observational period (Fig. 16b,d). In the low-resolution configurations, the simulated increase in freshwater content from the mid-1990s to the mid-2010s falls mostly within the uncertainty range of observational estimates (Fig. 16b). However, in the high-resolution configurations, the model-observation misfit becomes more pronounced in most models (Fig. 16d). The high-resolution CMCC-NEMO model shows a



**Figure 16.** (a) Time series of liquid freshwater content (FWC) in the Arctic basin in the low-resolution models. (b) The same as (a), but for the anomalies relative to the 1992-2008 mean. (c)(d) The same as (a)(b), but for the high-resolution models. The observational estimate (Wang et al., 2019) is shown in (b)(d).

persistent increase in freshwater content from the mid-1990s until the end of the simulation, contrary to observations indicating a leveling off in the mid-2010s (Wang et al., 2019). In contrast to high-resolution CMCC-NEMO, both high-resolution ACCESS-MOM and IAP-LICOM models simulate a declining trend starting from the early 2010s, which differs from the observed leveling off in the mid-2010s. Only AWI-FESOM and FSU-HYCOM reproduce the leveling off of freshwater content in the mid-2010s in the high-resolution models. FSU-HYCOM performs the best in simulating the temporal changes in freshwater content, as both of its configurations produce freshwater content anomalies that fall within the observational uncertainty range.

Several factors can influence Arctic freshwater content, such as winds, sea ice effects on momentum transfer, and the surface geostrophic currents which influence the circulation pathway and residence time of freshwater in the Arctic Ocean (Wang et al., 2021). The two models that show the greatest deterioration in simulating freshwater content changes in their high-





resolution configurations compared to their low-resolution configurations, ACCESS-MOM and CMCC-NEMO, exhibit the largest biases in surface salinity among the models (Fig. 7). ACCESS-MOM has limited sea surface salinity restoring, and it is switched off under sea ice in CMCC-NEMO. These findings suggest that model resolution is not the dominant factor influencing the model's performance in simulating the mean state of freshwater spatial distribution and the temporal changes in Arctic freshwater content. The models tend to need sea surface salinity restoring to climatology to avoid large salinity biases at surface.

## 3.5 Gateway transports

Arctic climate is strongly influenced by inflows from the Atlantic and Pacific oceans. As mentioned in Section 3.4, the transport of ocean heat from lower latitudes significantly affects the temperature of the Arctic Ocean (Polyakov et al., 2020; Shu et al., 2022), extent of Arctic sea ice (Woodgate et al., 2010; Årthun et al., 2012; Shu et al., 2021; Pan et al., 2023), and winter air temperature (Screen and Simmonds, 2010; Nummelin et al., 2017). The Arctic Ocean also exports freshwater to the subpolar North Atlantic, with potential impacts on upper ocean stratification, deep water formation, large-scale circulation, and climate dynamics (Aagaard et al., 1985; Goosse et al., 1997; Arzel et al., 2008). Furthermore, the inflows and outflows through the Arctic Ocean gateways play a crucial role in the transport of nutrients and planktonic organisms (Walsh et al., 1989; Hátún et al., 2017; Basedow et al., 2018). Observations and model simulations consistently indicate that ocean heat convergence to the Arctic Ocean and the hydrological cycle in the Arctic region are intensifying under a warming climate (Wang et al., 2023). In this subsection, we will assess the models' ability to simulate the mean state and temporal changes in Arctic-Subarctic ocean transports through key gateways (the Bering Strait, Barents Sea Opening, Fram Strait and Davis Strait) shown in Fig. 1.

The ocean volume ($VT$), heat ($HT$), and freshwater ($FWT$) transports through a gateway transect are defined as follows:

$$VT = \iint u_n \mathrm{d}z \mathrm{d}\ell \tag{2}$$

$$HT = \iint \rho_o c_p u_n (\theta - \theta_{ref}) \mathrm{d}z \mathrm{d}\ell, \tag{3}$$

$$FWT = \iint u_n (S_{ref} - S)/S_{ref} \mathrm{d}z \mathrm{d}\ell, \tag{4}$$

where $u_n$ represents the ocean velocity perpendicular to the transect, $\theta$ denotes potential temperature, $\theta_{ref}$ is the reference temperature, $S$ indicates salinity, $S_{ref}$ is the reference salinity, $\rho_o$ corresponds to ocean density, $c_p$ represents the specific heat capacity of seawater, and the integration is performed over the height $z$ from the ocean bottom to the surface and over the distance $\ell$ along the transect. Ocean heat transports are calculated relative to $\theta_{ref} = 0^\circ$C, and freshwater transports are calculated relative to $S_{ref} = 34.8\,\mathrm{psu}$, which is an estimate of the mean salinity of the Arctic Ocean (Aagaard and Carmack, 1989).



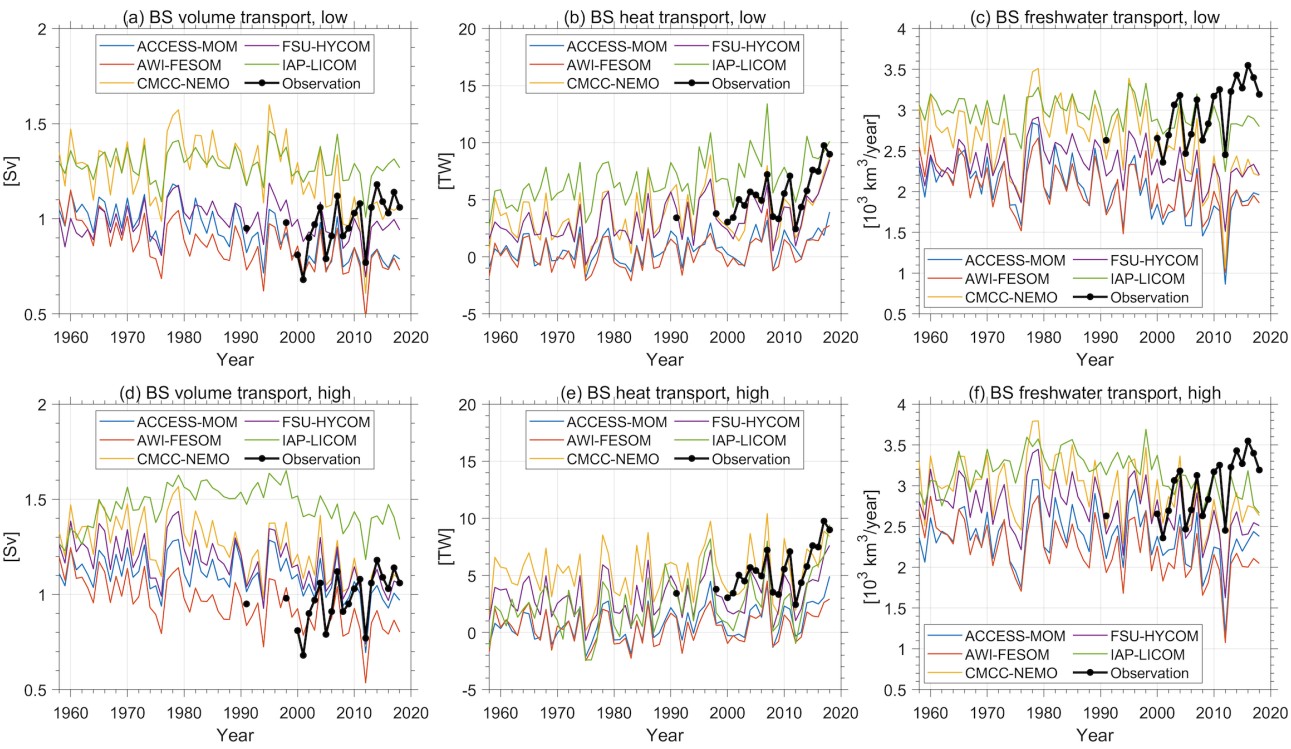

**Figure 17.** Time series of ocean (a) volume, (b) heat and (c) freshwater transports in the Bering Strait (BS) in low-resolution models. (d)(e)(f) The same as (a)(b)(c), but for high-resolution models. Heat transport is referenced to $0°C$, and freshwater transport is referenced to 34.8 psu. The observational estimates are adopted from Woodgate and Peralta-Ferriz (2021).

Monthly velocity, temperature and salinity data are available from the model outputs and used in the calculations so eddy transports are largely neglected. It was suggested that heat directly transported by eddies is small at the Fram Strait (Kawasaki and Hasumi, 2016), while eddies can influence the mean flow into the Arctic basin by altering the distribution of the Atlantic Water current between the re-circulation branch and the inflow branch (Wekerle et al., 2017; Hattermann et al., 2016). Additionally, it should be noted that mooring instruments used for measuring ocean transports have low spatial resolutions without
covering whole gateway transects, and as a result, the uncertainties associated with transport estimates are usually significant (e.g., Beszczynska-Moeller et al., 2011; Wang et al., 2023). Nonetheless, despite these limitations, these estimates represent the most reliable data currently available for evaluating models.

### 3.5.1    Bering Strait

The Bering Strait volume transport had a climatological value of $0.8\pm0.2\,\mathrm{Sv}$, but it increased to $1\pm0.1\,\mathrm{Sv}$ in the last two decades
(Woodgate and Peralta-Ferriz, 2021). Both the ocean heat and freshwater transports also increased during this period, from $4\,\mathrm{TW}$ and $2400\pm300\,\mathrm{km}^3/\mathrm{year}$ in 1980–2000 to $6\,\mathrm{TW}$ and $3000\pm280\,\mathrm{km}^3/\mathrm{year}$ in 2000–2020 (Woodgate and Peralta-Ferriz,





2021; Wang et al., 2023). The low- and high-resolution models exhibit similar spreads in the Bering Strait volume, heat, and freshwater transports (Fig. 17). Despite the model spreads, the interannual variability of the Bering Strait transports is highly consistent among the models regardless of model resolution (Figure S5), as observed in previous model intercomparisons (Wang et al., 2016a; Shu et al., 2023).

It has been observed that low-resolution ocean models struggle to reproduce the observed upward trend in Bering Strait volume transport (Shu et al., 2023). Increasing the resolution does not improve this issue in any of the models analyzed in our study (Fig. 17a,d). As these models employ various numerical methods, resolutions, and parameterizations, but still exhibit the same issue, it is likely that the problem originates from the atmospheric reanalysis data (JRA55-do) used to drive these models. The models are able to capture the observed increase in heat transport over the past decade (Fig. 17b,e), indicating that the warming of the Pacific Water contributes partially to the increase in ocean heat transport. However, none of the models simulate the observed increase in freshwater transport (Fig. 17c,f), primarily because the increase in volume transport significantly contributes to the rise in freshwater transport (Woodgate and Peralta-Ferriz, 2021). Overall, for the Bering Strait, both the model spreads and the models' ability to simulate interannual variability and decadal trends are not substantially influenced by model resolution.

### 3.5.2 Barents Sea Opening

The ocean volume transport through the Barents Sea Opening did not show a statistically significant trend over the past few decades, but the ocean heat transport exhibited an upward trend (Skagseth et al., 2020). Based on mooring observations in the 1990s and 2000s, the climatology of ocean volume transport is estimated to be between 2 and 2.3 Sv (Smedsrud et al., 2010, 2013). The models tend to overestimate the volume transport in both their low-resolution and high-resolution configurations (Fig. 18a,d). The low-resolution CMCC-NEMO model stands out as an outlier, with a volume transport nearly twice that of the observations, while this bias is reduced in its high-resolution counterpart. The heat transport in the Barents Sea Opening was approximately 70 TW in the 2000s (Smedsrud et al., 2013). Two low-resolution models, FSU-HYCOM and IAP-LICOM, underestimate the heat transport, while their high-resolution counterparts exhibit higher heat transport, becoming similar (IAP-LICOM) or even larger (FSU-HYCOM) than the observations (Fig. 18b,e). Although increasing the horizontal resolution improves the ocean volume transport in CMCC-NEMO, the high-resolution model still exhibits a high bias in heat transport, indicating the influence of warmer ocean temperatures.

The model spreads in the Barents Sea Opening volume and heat transports are slightly reduced in the high-resolution models (Fig. 18a,b,d,e). The interannual variability of ocean volume and heat transports is consistent among the models and is not strongly influenced by model resolution (Figure S6). A synthesis of models and observations suggests an increase in heat transport of approximately 8 TW from 1980–2000 to 2000–2020 (Wang et al., 2023). The models simulate a consistent trend with an increase close to this value.

The Atlantic Water inflow in the Barents Sea Opening is saltier than the average salinity of the Arctic Ocean, making it a freshwater sink for the Arctic Ocean. The net freshwater transport in the Barents Sea Opening is a small, negative value, estimated to be around $-100\,\mathrm{km^3/year}$ (Serreze et al., 2006). In both the low-resolution and high-resolution models, there



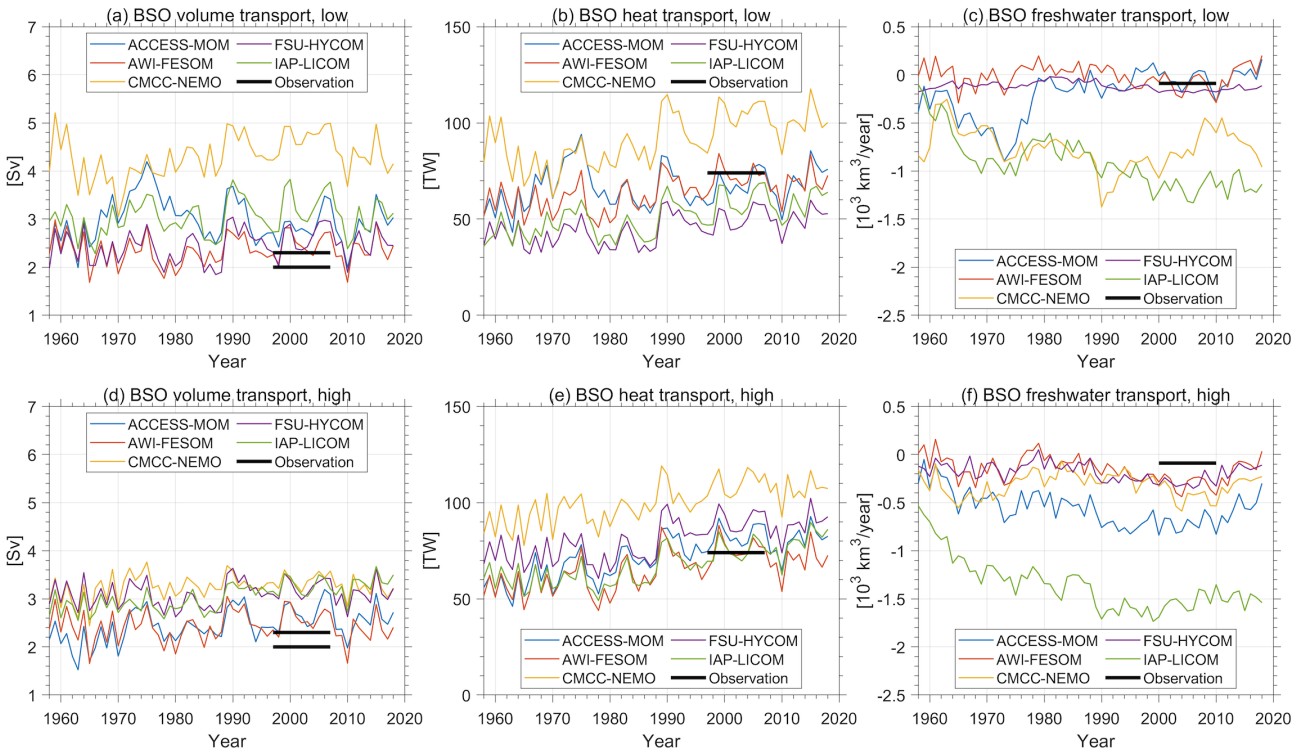

**Figure 18.** The same as Fig. 17, but for Barents Sea Opening (BSO). The observational estimates are taken from Smedsrud et al. (2013) and Serreze et al. (2006).

are two models that simulate excessively large negative values (Fig. 18c,f). IAP-LICOM exhibits the largest biases in both groups. As it does not have outlier volume transports, the biases in freshwater transport are primarily due to its positive salinity biases in the inflow. The interannual variability of freshwater transport is not consistent among the low-resolution models but improves in the high-resolution models (Figure S6).

### 3.5.3 Fram Strait

The climatological net volume transport through Fram Strait is estimated to be $-2 \pm 2.7$ Sv (Schauer et al., 2008). Among the low-resolution configurations, AWI-FESOM and FSU-HYCOM exhibit a good representation of the mean volume transport, while four of the high-resolution configurations perform well, except for CMCC-NEMO (Fig. 19a,d). The mean heat transport through Fram Strait was approximately 30 TW in the period 1980–2000 and increased to about 40 TW in 2000–2020 (Wang et al., 2023). Three of the low-resolution configurations (CMCC-NEMO, FSU-HYCOM, and IAP-LICOM) show insufficient heat transport (Fig. 19b), which contributes to their strong cold biases in the Atlantic Water layer (Fig. 4 and 5). In all the models, the heat transport increases with resolution (Fig. 19b,e), with the weakest increase observed in AWI-FESOM, possibly due to the same model resolution outside the Arctic Ocean in both configurations. CMCC-NEMO and FSU-HYCOM exhibit ex-



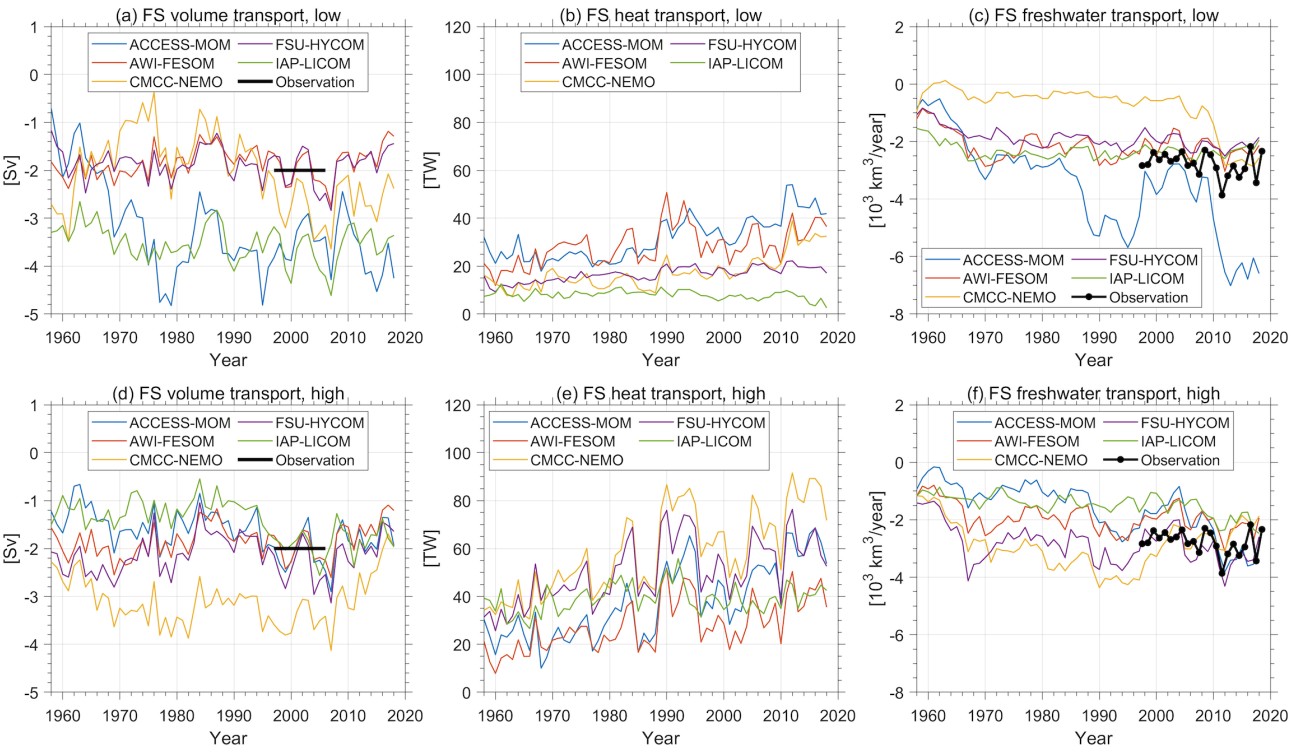

**Figure 19.** The same as Fig. 17, but for Fram Strait (FS). The observational estimates are taken from Schauer et al. (2004) and Karpouzoglou et al. (2022).

cessively high heat transport in their high-resolution configurations, contributing to the excessively warm Atlantic Water layer
in these models (Fig. 6h,i). The climatological freshwater transport in Fram Strait is approximately $-2700 \pm 530 \, \mathrm{km^3/year}$
(Serreze et al., 2006). Two low-resolution models, CMCC-NEMO and ACCESS-MOM, either significantly underestimate
or overestimate the freshwater transport in Fram Strait (Fig. 19c). The model spread in Fram Strait freshwater transport is
significantly reduced in the high-resolution models (Fig. 19f).

We observed that the simulated interannual variability of ocean volume, heat, and freshwater transports in Fram Strait
improves noticeably with increasing resolution (Fig. 19 and Figure S7). Most of the low-resolution models tend to exhibit weak
interannual variability in Fram Strait heat and freshwater transports. With the exception of IAP-LICOM, all the high-resolution
models simulate an increase in heat transport in the early 1990s and the first two decades of the 21st century, consistent with
the changes in observational implication and previous model studies (Polyakov et al., 2013; Wang et al., 2020b). Observations
indicate an increase in freshwater export in 2010–2013 compared to the 2000s, manifested by strengthened currents and lower
salinity (de Steur et al., 2018). All the high-resolution models simulate an increase in freshwater export over this period, with
two models (AWI-FESOM and FSU-HYCOM) even capturing a magnitude similar to the observed increase (Fig. 19f). In





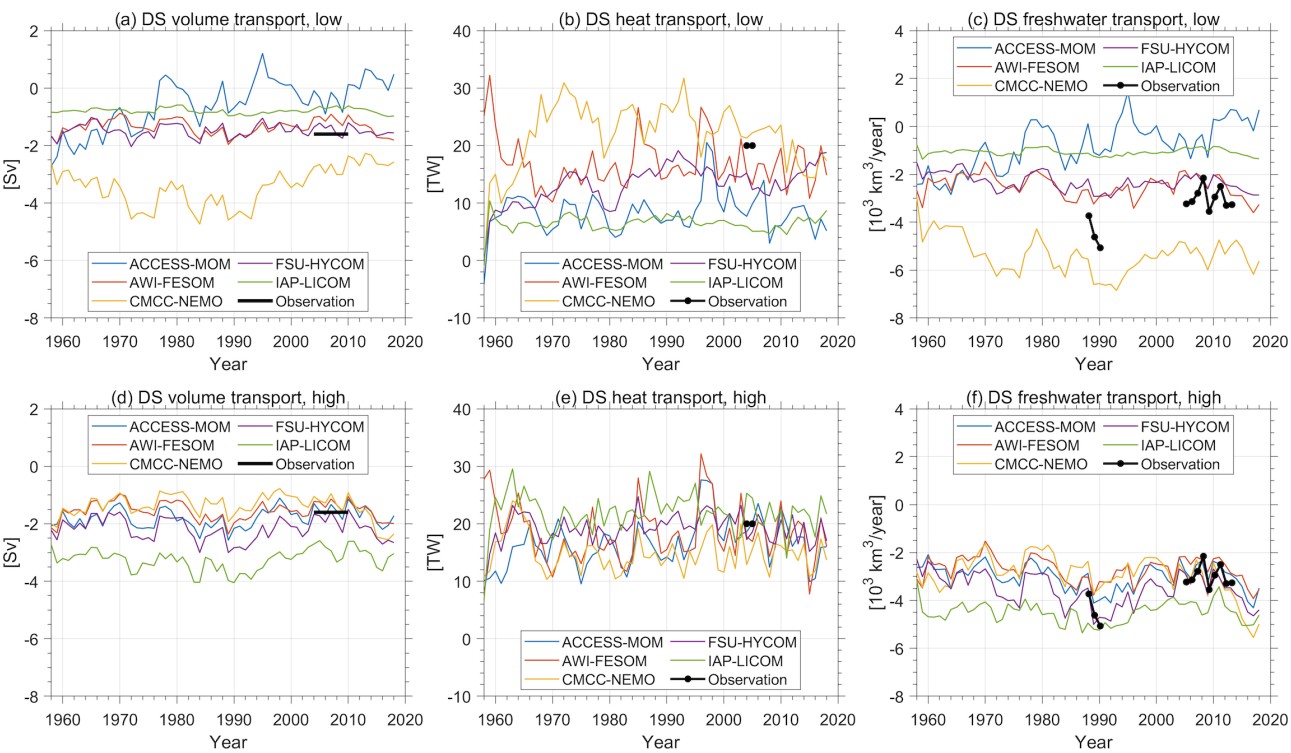

**Figure 20.** The same as Fig. 17, but for Davis Strait (DS). The observational estimates are taken from Cuny et al. (2005) and Curry et al. (2014).

contrast, all the low-resolution models exhibit either an overestimation or underestimation of the magnitude of this freshwater transport change (Fig. 19c).

### 3.5.4 Davis Strait

The volume transport in Davis Strait was estimated to be $-2.6 \pm 1$ Sv in 1987–1990 (Cuny et al., 2005) and $-1.6 \pm 0.5$ Sv in 2004–2010 (Curry et al., 2014). Among the low-resolution models, IAP-LICOM exhibits a low volume export without clear interannual variability (Fig. 20a) due to the closure of the western straits in the Canadian Arctic Archipelago (Fig. 2e). The low-resolution ACCESS-MOM model shows unrealistically positive volume transport (inflow to the Arctic) in some years. In both low-resolution IAP-LICOM and ACCESS-MOM models, the biases in volume transport in Davis Strait are anticorrelated

with the biases in Fram Strait (Fig. 19a and 20a) because the Arctic export is distributed between these two gateways (Wang et al., 2023). The low-resolution CMCC-NEMO model exhibits excessively high volume export, nearly double the observed values. The model spread in the mean volume transport is significantly reduced in the high-resolution models (Fig. 20d). The climatological heat and freshwater transports in Davis Strait are estimated to be $18 \pm 17$ TW and $-3200 \pm 320 \, \mathrm{km}^3/\mathrm{year}$ (Cuny et al., 2005), respectively. Similar to the biases in volume transport, the heat and freshwater transports in Davis Strait





are either too low or too high in the three aforementioned low-resolution models (Fig. 20b,c). Increasing resolution reduces the model spread and brings the results closer to the observations for both heat and freshwater transports (Fig. 20e,f). It is important to acknowledge that the observation of Davis Strait heat transport has a very limited time span and is accompanied by substantial uncertainty. However, irrespective of this limitation, a decrease in model spread suggests potential improvements in high-resolution models.

Increasing resolution clearly improves the inter-model consistency in the simulated interannual variability of ocean volume and freshwater transports in Davis Strait, but not for heat transport (Figure S8). This indicates that the models exhibit less agreement in simulating the advection of water of Atlantic origin from the Irminger Sea to Baffin Bay. The high-resolution models consistently simulate a reduction in Davis Strait volume and freshwater exports in the early 1990s and an increase in the mid-to-late 2010s. The former reduction is primarily due to the positive Arctic Oscillation, which shifted more Arctic export to Fram Strait, while the latter increase is mainly attributed to the drop in dynamic sea level south of Greenland (Wang et al., 2022a, 2023). The increase in Davis Strait freshwater export from 2010 to 2017 exceeded $1500\,\mathrm{km^3/year}$ as suggested in a previous modeling study (Wang et al., 2022a). This magnitude of increase is quantitatively reproduced in all the high-resolution models except for CMCC-NEMO, which simulates an excessively high increase.

## 4   Conclusion and Discussion

This paper assesses Arctic Ocean simulations using five pairs of matched low- and high-resolution models within the CMIP6 OMIP-2 framework (Griffies et al., 2016). The primary objective is to investigate whether increasing resolution can mitigate the typical model biases in low-resolution models identified in previous studies, which have persisted for more than two decades. The low resolution ($1°$ to $1/4°$) represents CMIP6 conditions, while the high resolution ($1/10°$ or better) represents future CMIP conditions. The main findings are summarized below.

1. An increase in horizontal resolution helps reduce biases in mean temperature and salinity. The low-resolution models exhibit warm biases below the core depth range of the Atlantic Water layer, even when cold biases are present in the Atlantic Water layer. This reflects the common issue of the Atlantic Water layer being excessively thick in low-resolution models. Additionally, the halocline and upper Atlantic Water layer exhibit biased fresh conditions in low-resolution models. These issues have been linked to spurious vertical mixing (Holloway et al., 2007; Wang et al., 2016a). Improved horizontal resolution alleviates these issues in some model pairs, but not in all cases, as differences in parameterizations, vertical resolution, and sea ice dynamics within each model pair could also influence the comparison. Three of the low-resolution configurations display significant cold biases in the Atlantic Water layer, which can be attributed to insufficient warm Atlantic Water inflow in the Fram Strait and excessive cold water originating from the Barents Sea, as found in previous analyses of low-resolution models (Ilicak et al., 2016; Shu et al., 2023). Higher resolution reduces the cold biases in all these models by enhancing Fram Strait heat import and reducing the cold bias in the northeastern Barents Sea.





2. Decadal warming events in the Atlantic Water layer are better simulated with higher resolution. While only one low-resolution model adequately reproduces the warming of the Atlantic Water layer in the 1990s and 2010s, four high-resolution models can do so. These warming events arise from episodes of intensified heat flux through the Fram Strait, which are more accurately represented in high-resolution models.

3. High-resolution models exhibit shallower surface MLD, reflecting the influence of permitted but not well-resolved eddies in restratifying the mixed layer. Two low-resolution models significantly overestimate the MLD, but this bias is reduced in their high-resolution counterparts in parts of the Arctic basin. However, the reduction in the MLD bias is not accompanied by a reduction in salinity bias; instead, it is associated with a distinct salinity bias. Among the models capable of reasonably simulating the warm Atlantic Water layer (one low-resolution and four high-resolution models), the shoaling trend of the cold halocline base depth in the eastern Eurasian Basin during the 2010s is captured. However, the high-resolution models do not consistently simulate changes in the cold halocline base depth on multi-decadal timescales.

4. Model performance in simulating the mean state of freshwater spatial distribution and the temporal changes in Arctic freshwater content does not improve with higher resolution. Although the bias in halocline salinity is reduced in high-resolution models, the sea surface salinity bias could worsen depending on the approach used for sea surface salinity restoring. This factor appears to have a non-negligible impact on the model's representation of Arctic freshwater content.

5. An increase in horizontal resolution improves the simulation of Arctic gateway transports, primarily for the Fram and Davis straits. For these two gateways, high-resolution models exhibit reduced spreads in the transports, closer agreement with observations regarding the mean states, and improved quantitative representation of variability and changes. Models agree more on the temporal variability than the mean state of the gateway transports as found in previous model intercomparison studies (Wang et al., 2016a; Shu et al., 2023). Increasing resolution does not resolve the challenge of simulating the observed increase in Pacific Water inflow in the 2010s, suggesting that the origin of this issue may lie in the common atmospheric forcing.

Overall, the use of higher horizontal resolution improves the model representation of the Arctic Ocean, although not all models achieve improvements for all climate-relevant variables. The improvements in Arctic temperature and salinity can be mainly attributed to enhanced warm inflow through the Fram Strait, more realistic watermass transformation in the Barents Sea, and decreased vertical mixing. The decreased mixing is evident in the reduction of warm biases at depth and fresh biases in the halolcine in the Eurasian Basin in some of the high-resolution models.

Interestingly, the low-resolution AWI-FESOM model (with a horizontal resolution of 24 km inside the Arctic Ocean, which is equivalent to nominal resolution of about 1/2° on tripolar grids, and 47 z-levels) exhibits more realistic hydrography and stratification than some of the high-resolution models (with a horizontal resolution of 1/10° or better, and similar or more vertical levels). Therefore, in future ocean model development for improving Arctic Ocean simulation in subsequent phases of CMIP, tuning model parameterizations and/or some numerical aspects is just as crucial as increasing model resolution.





However, for most metrics, a majority (typically 4 out of 5) of high-resolution models are in better agreement with one another and observations than their low-resolution counterpart, so what made AWI-FESOM successful at both low and high resolutions are not a general requirement for high-resolution simulation skill.

Our model intercomparison offers some clues for model developers and users to improve certain processes in their models. For instance, the low-resolution ACCESS-MOM model (with a horizontal resolution of about 9 km, higher than in the other low-resolution models) exhibits the highest net heat transport through the Fram Strait among the low-resolution models, but it has a significant cold bias in the Arctic basin. This suggests that the cold water originating from the Barents Sea is the main cause of the basin's cold bias. The other two low-resolution models (CMCC-NEMO and FSU-HYCOM) with cold biases in the Arctic basin also simulate excessively cold water in the northeast Barents Sea. In the high-resolution configurations of these models, the cold biases in the northeast Barents Sea are largely eliminated. However, as the other two low-resolution models do not exhibit significant cold biases in the northeast Barents Sea, it is possible that low resolution alone is not the primary cause of the cold biases. Previous analyses of forced ocean-ice models (Ilicak et al., 2016) and coupled climate models (Shu et al., 2019) have actually shown that some models could have overly warm water originating from the Barents Sea. Therefore, investigating the air-sea heat exchange and water mass transformation in the Barents Sea within low-resolution models exhibiting strong cold or warm biases may offer insights into effectively reducing Arctic temperature biases in low-resolution models. The fact that increasing resolution does help reduce cold biases in the northeast Barents Sea in our analyzed models implies that some resolution-dependent parameterizations or numerics in these models may be responsible for the biases.

Two of the models included in this study allow us to clearly distinguish the impacts of horizontal resolution from vertical resolution. In FSU-HYCOM, the high-resolution configuration has coarser vertical resolution compared to the low-resolution configuration. Therefore, the improved simulation of Atlantic Water layer temperature, halocline salinity, and some gateway transports in high-resolution FSU-HYCOM cannot be attributed to better vertical resolution. On the other hand, in AWI-FESOM, the vertical resolution remains the same in both configurations. The reduced thickness of the Atlantic Water layer and improved cyclonic circulation of the Atlantic Water in the deep basin in the high-resolution configuration are therefore associated with increased horizontal resolution. Among the five model pairs, AWI-FESOM exhibits the smallest difference between the two configurations. Firstly, its low-resolution configuration does not exhibit extreme biases, leaving less room for improvement. Secondly, the resolution outside the Arctic is the same in both AWI-FESOM configurations, indicating that the difference in simulation results is solely due to the local differences in model configurations. In other models, the impacts of different resolutions outside the Arctic can propagate into the Arctic Ocean through gateway transports, contributing to the Arctic differences within the model pairs.

In terms of the model representation of Arctic freshwater, particularly regarding spatial distribution and temporal changes in freshwater content, significant improvement is not observed in high-resolution configurations. Although salinity biases in the halocline are reduced in some high-resolution configurations, two models exhibit similar or even larger sea surface salinity biases when their spatial resolution is improved. This could potentially be attributed to weak or absent sea surface salinity restoring. The strong dependence of simulated sea surface salinity on numerical restoring indicates the general need for improvements in the surface freshwater budget and the processes influencing freshwater circulation, such as the impact



of sea ice on momentum transfer in models. Reducing the overall biases in salinity, which remain large in most of the high-resolution models, is crucial, as the surface geostrophic currents in the Arctic are directly influenced by the spatial pattern of the freshwater column (Armitage et al., 2017; Wang, 2021).

It is worth noting that the high-resolution OMIP-2 models assessed in this study only marginally permit mesoscale eddies in the Arctic Ocean. The influence of eddies appears to be reflected in the disparity in winter MLD between two configurations for 655 some of the models. However, there is no conclusive evidence suggesting that the major improvements in the high-resolution models are attributed to simulated eddies in the Arctic Ocean. In particular, the first baroclinic Rossby radius in the Barents Sea is extremely small (approximately 2 km; Nurser and Bacon, 2014) and these high-resolution models cannot adequately resolve eddies in this region. Therefore, the reduction in the large temperature bias in the northeastern Barents Sea and the Arctic deep basin in three of the analyzed models cannot be attributed to resolved eddies in the Barents Sea. Moreover, despite 660 that eddy transport was proposed to be one of the key factors influencing the amount of freshwater in the Beaufort Gyre (Manucharyan and Spall, 2016; Meneghello et al., 2017), we did not observe significant improvements in the simulated mean state or variability of the Arctic freshwater content in the high-resolution models.

While sea ice properties have not been a focus of this paper, Arctic and Antarctic sea ice concentrations and thicknesses in March and September are discussed in the same model pairs (Chassignet et al., 2020) and/or in the corresponding model 665 description papers cited in Section 2. To summarize previous findings briefly, March Arctic sea ice concentration fields were similar among the models at all resolutions, and September Arctic sea ice concentration fields were more sensitive to models than to spatial resolution. The AWI-FESOM model had the most realistic September sea ice conditions at both low and high resolution, consistent with the findings about Arctic watermass properties here. Sea ice thicknesses differed significantly among the models in all seasons.

It is unlikely that most climate models participating in near-future CMIP7 deck and scenario simulations will have resolutions higher than those employed in the current high-resolution OMIP-2 models. Therefore, our evaluation of the OMIP-2 models provides valuable and timely information for groups preparing future CMIP simulations. In particular, we suggest that some of the extreme model biases are not primarily due to low resolution alone, and investigating model numerics and parameterizations, including the practical treatment of narrow straits in the ocean, could potentially improve the representation of the 675 Arctic Ocean in medium-resolution models used in climate-scale simulations.

*Code and data availability.* The following OMIP model output, published on the Earth System Grid Federation, has been used: ACCESS-MOM (ACCESS-OM2) (Holmes et al., 2021), CMCC-NEMO (CMCC-CM2-SR5) (Fogli et al., 2020), IAP-LICOM (FGOALS-f3-H and FGOALS-f3-L) (Lin, 2019, 2020). The 1/10° ACCESS-MOM data is available from http://dx.doi.org/10.25914/608097cb3433f. The model data used to produce the figures and the corresponding analysis scripts are archived at http://dx.doi.org/10.5281/zenodo.8046638.



*Author contributions.* QW coordinated the conceptualization and wrote the first draft of the manuscript. QS and SW processed the model data and produced the figures. QW, AB, EC, PF, AH, DI, AK, NK, YL, PL, HL, PS, DS and XX provided model datasets and expertise for the interpretation of the results. IP provided the gridded AWCT and halocline observational data. All authors contributed to the interpretation of the results, discussion of the scientific content, and improvement of the manuscript.

*Competing interests.* Qiang Wang is a member of the editorial board of Geoscientific Model Development.

*Acknowledgements.* We thank the present and past members of the CLIVAR Ocean Model Development Panel who have designed and supported the Ocean Model Intercomparison Project (OMIP). QW is supported by the Helmholtz Climate Initiative REKLIM (Regional Climate Change and Human) and the EPICA project in the research theme "MARE:N – Polarforschung/MOSAiC" funded by the German Federal Ministry for Education and Research with funding no. 03F0889A. QS is supported by the National Natural Science Foundation of China (grant no. 42276253), the Shandong Provincial Natural Science Foundation (grant no. ZR2022JQ17), and the Taishan Scholars Program
(grant no. tsqn202211264). AK is supported by ARC grants LP200100406. BFK is supported by the SASIP project of the Schmidt Futures Foundation. The ACCESS-MOM model runs were undertaken with the assistance of resources from the National Computational Infrastructure (NCI) and the Consortium for Ocean-Sea Ice Modelling in Australia (COSIMA; http://www.cosima.org.au), which are supported by the Australian Government.





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
