# Peer review of "Impact of increased resolution on Arctic Ocean simulations in Ocean Model Intercomparison Project phase 2 (OMIP-2)"

_Geoscientific Model Development, 2023_

## Referee Comment (RC2)

Review of gmd-2023-123
**Title:** Impact of high resolution on Arctic Ocean simulations in Ocean Model Intercomparison Project phase 2 (OMIP-2)

**General comments**

The topic of the study is very interesting and important. The ocean component of current coarse resolution climate models (CMIP6) has large biases in the Arctic Ocean, and future climate projections of the hydrography in the Arctic Ocean show a large model spread. Thus, it is necessary to investigate approaches that can reduce the model biases, such as using higher resolution versions of the ocean components. This is the focus of this study, to investigate how higher resolution versions improves the hydrography and ocean circulation in the Arctic Ocean compared to lower resolution versions. The authors use 5 different ocean components with realistic atmospheric forcing, where each component has a high and a low resolution version.

The manuscript is overall nicely written and has very interesting results with good figures. However, there are two major comments: **(1)** Some discussions points are missing in the present study, as listed below. Although discussion points, I think it is important to discuss some of these points early in the manuscript, perhaps as part of the approach used in the study (i.e., in Section 2). **(2)** The structure of Section 4 (Conclusion and Discussion) is to some extent difficult to follow. I therefore recommend major revision. I believe the authors will be able to improve on these two points. In addition, there are parts of the manuscript text that could be clarified and/or improved. I've pointed to some of these places further below (see 'Comments per section').

**(1) The manuscript needs discussion:**
- On how representative the 5 ocean models are compared to other CMIP6 ocean models (Shu et al., 2023). It seems that some of the 5 ocean models used in this study have poor performance compared to other CMIP6 ocean models.
- On the very different sets of resolution for each model. For instance, one ocean model has 9km as low resolution and 3.6km as high resolution, whereas another model has 72km as low resolution and 6.8km as high resolution.
- On the impact of increased horizontal resolution when the vertical resolution is changing at the same time (in some cases the parameterization or sea ice model also changes when moving from low to high resolution version). This is to some extent discussed in Section 4.
- On the deep-water balance, as the spin-up does not appear to be long enough.
- On the benefits of high resolution version when the low resolution has already good performance.*

*The study shows that the high resolution versions outperforms the low resolution versions when the latter is showing poor performance (3 out of 5 models, where the low resolutions are 9km, 32km, and 51km). But when the low resolution has good performance, the benefits of the high resolution are not that clear (AWI-FESOM). In addition, one model shows that higher resolution does not necessarily lead to improvements (IAP-LICOM).
*The good performance of the low resolution of AWI-FESOM is mentioned in Section 4 (lines 610-612), and the discussion on lines 613-614 is very interesting. Could you please expand on this discussion; why is low resolution AWI-FESOM working quite well despite its coarse resolution?

**(2) The structure and message of Section 4 needs to be clearer.**
- Section 4 is, as it stands now, sometimes repetitive and needs more structure.
- The results could be more concrete; the results are sometimes a bit vague. For instance, how much is the performance improved by increasing the horizontal resolution? Could it be quantified?
- The results seem to be sometimes simplified, e.g., line 570. Could the discussion be more nuanced? For instance, one model shows good performance for both versions, one model shows overall poor results, whereas three models show poor results for low resolution but improved for the high resolution.
- As the five different models shows quite different results (and the configurations are different from model to model), it might be useful to discuss each of the models separately.
- I think it would benefit the manuscript to split Section 4 into two sections; first Discussion and thereafter summarize and conclude the study in Conclusion.

**Comments per section**

**Abstract**
Line 6: The main improvements appear to be seen in the Fram Strait and Davis Strait, but not much improvement seems to be the case in the Bering Strait. Suggest making the 'Arctic gateways' more nuanced.

**Introduction**
Line 15: Note that the amplified Arctic Ocean warming compared to global is for the future, or is the 'two times faster' referring to present?

Please clarify which models are ocean components forced with realistic atmosphere, coupled, regional or global models.

**Results**
Line 168-169: How does this relate to Heuze et al. (2023; this study is already in your reference list) that finds that the ocean heat transport through the Fram Strait is reasonable, but the volume transport and temperature is not correct?

Line 175: The recent study by Richards et al. (2022) might be useful here.

Line 172-180: Here the boundary currents are discussed. Have you looked into the ocean circulation/boundary currents in the models?

Line 184-185: This sentence needs a reference. Could there be cold sources in the deep also entering through the Fram Strait, such as Norwegian Sea Deep Water?

Line 192: The results described here are difficult to see from the figures.

Line 198: It is not fully clear how the absence of the sea ice dynamics impacts the results.

Line 219: The discussion on how the sea surface salinity restoring impacts the results is not fully clear to me.

Line 225: A reference is needed.

Line 254: Poleward propagation of warm anomalies in the Atlantic Water layer is discussed in Årthun et al. (2017).

Line 259: How is sea ice decline leading to intensified inflow?

Line 260: Please be consistent, use either Arctic basin or Arctic Basin.

Line 270-272: This is not so easy to see from the figures.

Line 280: Please explain why these two periods are compared.

Line 349: Note that a different approach is taken in Muilwijk et al. (2023) studying the halocline in CMIP6 models.

Line 363: Is it the deepening of the Atlantic Water layer that causes the deepening of the halocline, or opposite, or other factors causing the deepening of the layers?

Line 368: Should I expect to see that 1990s is warmer than 2010s for FSU-HYCOM? Please clarify.

Line 417: A positive salinity bias both in the Amerasian Basin and the Eurasian Basin?

**Conclusion and Discussion**
Please see the comments above regarding Section 4.

Several places in this section it is mentioned that higher resolution helps, but at the same time other findings reveal that the answer is not that straightforward (for instance AWI-FESOM and IAP-LICOM show different results than the three other models). This comment is similar to my comment above, about giving a bit more nuanced discussion.

Line 629: The results from Pan et al. (2023) are relevant here.

*Figures*
**Fig. 14:** The dark red color in figure 14j is very difficult to see. Perhaps write a note in the caption that the Amerasian Basin is deep red in 14j.

**References:**
Århun, M., Eldevik, T., Viste, E. *et al.* Skillful prediction of northern climate provided by the ocean. *Nat Commun* 8, 15875 (2017). https://doi.org/10.1038/ncomms15875

Richards, A. E., Johnson, H. L., & Lique, C. (2022). Spatial and temporal variability of Atlantic Water in the Arctic from 40 years of observations. Journal of Geophysical Research: Oceans, 127, e2021JC018358. https://doi.org/10.1029/2021JC018358

---

## Author Comment (AC3)

**Reply to Reviewer #1**

Dear Reviewer,

Thank you very much for your very helpful comments. We revised the paper following your comments. See our reply below (in blue).

This is a fairly comprehensive investigation of an important topic that is overall well presented, especially for a challenging topic to present like Arctic Ocean modeling. The conclusions are clear, but could be quantified a bit more (see major comment below); I recommend publishing this manuscript after revisions.

My largest comment is about how the results are quantified. The individual model biases are quantified, but the improvement or lack thereof between low and high resolution simulations is not, and in most cases is only qualitatively described. Is there a way to quantify how much a simulation improves for a given metric? The statements similar to '4 out of 5 models improved' are technically quantitative, but I think going beyond this is worth the effort. A few specific examples:

- Figures 3-5 and 8 in particular should be better quantified. Why not plot Figure 3 as a bias the way Figure 7 is? Figure 7 explicitly plots the bias, and is an example where the improvement from low to high resolution is more obvious and quantifiable.

Reply: we calculated RMSE and displayed it in each panel of Figs. 3,4,5,7,8 to better quantify the model biases, and revised the text related to these figures. In both Fig. 3 and 7, now we show the profiles instead of model biases; this better illustrates how the temperature and salinity vertical profiles really look like. In the final conclusion section, we also provided root-mean-square errors for temperature and salinity averaged over the five models (L695-713).

- Figure 11 and lines 278-295: It's difficult to tell quantitatively which simulations agree best with the observations. What is the magnitude of the improvement (e.g., 100% error to 50% error)? Only a suggestion: can a basin-wide average warming be added in as a number on each sub-plot or as a separate table?

Reply: We added basin-mean values in each respective panel. In low-resolution models, two have negative mean values (cooling on average), inconsistent with observations. In high-resolution models, all have positive mean values, but three of them overestimate the observed decadal warming.

- Lines 402-403: CMCC-NEMO, quantitatively, which is better the low or high resolution case? Is there still a small improvement for higher resolution?

Reply: the freshwater column is underestimated in one version and overestimated in the other. So, increasing resolution does not improve its representation. There is no significant improvement for this metric in the models in general.

- The introduction is also not quantitative in terms of past studies. At some point it doesn't matter if there is 100% error or 200% error because it's wrong either way, but I was left wondering exactly how bad simulations are (apologies for being pessimistic).

Reply: We added "The model spread (standard deviation among models) of the Atlantic Water layer temperature reaches about 1°C and the multi-model-mean thickness of the Atlantic Water layer exceeds twice the observed value." (L34-36)

- Overall, there is a lot of time spent discussing the individual models and not enough time spent discussing the improvements. It is clear that the models need to be discussed individually and the errors pointed out, but can the balance be shifted a bit more to discussing potential improvements?

Reply: By incorporating RMSE information, we improved the overall quantification of model performance. However, a challenge remains as the models do not consistently show improvement in some metrics. In such instances, it is crucial to discuss the specific aspects that have seen improvement or not within individual models. In the revised main text, we slightly reduced the instances where individual model names are mentioned. In the revised conclusion section, we provided temperature and salinity RMSE averaged over the five models.

**Additional comments**

- Sea ice: How does increasing resolution effect the representation of sea ice? Better representation of leads or individual floes? Has this been addressed in other papers? Lines 663 – 669 about sea ice should be earlier in the paper, prior to the results section.

Reply: We moved the original text in the conclusion section to the end of section 2, and also briefly mentioned sea ice simulation in the previous OMIP study (L64-65, L145-150). According to previous studies, increasing resolution does not remarkably improve sea ice concentration and thickness.

- Line 75-81: forcing is from 1958 to 2018, and this is repeated 5 times (so 300 years). By only analyzing the first 60 years, do you expect the Atlantic water to be stable / accurate? Is there one of the 5 pairs where the first and last cycles could be compared for AW properties?

Reply: We added discussions about model integration length and spinup in the revised Discussion section. In particular, we analyzed temperature in one of the models which has a few simulation cycles. (L593-606)

Why these metrics? Are these the metrics that can be easily tested? What other metrics would be good to have but can't be compared with observations? Some introduction to this as a methods section or at the beginning of the results section would be appreciated. I also suggest explicitly

stating that high-resolution metrics are not considered because the focus is on improvements from low to high resolution.

Reply: The choice of the metrics are motivated by their climate relevance. We provided some concise background for each of the metrics at the beginning of each subsection (except for T/S), which serves as motivation. We also explicitly state that "Assessing mesoscale eddy activity in the high-resolution simulations is beyond the scope of this paper. It is worth noting that the high-resolution OMIP-2 models assessed in this study only marginally permit mesoscale eddies in the Arctic Ocean…" in the Discussion section.

- Surface circulation has been mentioned a few times in the paper (e.g., lines 407-408), but it is not explicitly considered. Could it be included, maybe by comparing to Armitage et al. circulation products (even if this is a different year grouping than the PCH climatology)?

Reply: We added the sea surface height plots with comparison to the satellite observation (Fig. S5). The impact of model resolution on sea surface height is similar to that on the freshwater column as expected. (L429-433)

- Mixed layer depth comparison: Consider switching Figure S3 and Figure 12. Lines 325 – 327 suggest that the Figure 12 comparison to observations is not valid and certainly has fewer cautionary remarks than the Schmidtko comparison in Figure S3.

Reply: in both figures, we used monthly model data to compute MLD, so Figure S3 is not better. We used different density criteria in these two figures. The criterion used in Figure 12 has been suggested to be more valid for the Arctic Ocean based on observations, so we put this figure in the main text.

- Conclusions: To me it seems that conclusions #1,2, and 5 all boil down to gateway transports, and these metrics related to gateway transports are improved in high-resolution simulations. Conclusions #3-4 are about the ability to represent salinity accurately, and high-resolution does not always help with this. Is this an oversimplification of the findings? I might leap to saying that parameterizations of gateway transports should be the focus in terms of improving low-resolution models. Yes?

Reply: We suggested that at least a few key factors are responsible for the identified model issues, including gateway transports, numerical mixing, processes influencing temperature in northeastern Barents Sea. We hope the revised Discussion section and Conclusion section can better provide the main information.

**Minor comments**

- Title: 'higher resolution' or 'increased resolution' seem less awkward to me.

Reply: we changed "high" to "increased"

- Line 89: 'updation'?

Reply: changed to "updates" (L96)

- Line 183: This was confusing. Maybe clarify that intermediate and deep layers refer to the water beneath the 0°C AW lower boundary.

Reply: we added "located below the lower 0°C isotherm" L201

- Line 212: Primarily in the Eurasian Basin. Only improvement for the Amerasian Basin is NEMO, and ever so slightly HYCOM.

Reply: we changed "particularly" to "primarily" L228

- Figures 4, 8: Why is 400 m depth chosen as the depth to investigate geographic patterns?

Reply: The 400 m depth was traditionally chosen to illustrate the Atlantic Water layer (e.g., Ilicak et al., 2016; Shu et al., 2023), although the core depth of this layer changes in space. Recognizing the incompleteness in showing the information by fixing the depth, we have shown the AWCT in Fig. 5.

- Line 250: is there a way to reconstruct what the total mixing from all sources is for these models (explicit, applied parameterizations, and numerical)? How different would the total mixing be?

Reply: this would need a big effort from all individual model groups, which is not possible to realize for this paper. The mixing coefficients computed from mixing schemes are often not in the variable list for model output, and diagnosing the numerical mixing needs specific techniques. In the paper we can only discuss the impact of vertical mixing by combining what we know from literature and the simulated changes in the temperature and salinity in the analyzed models.

- Line 270-272: Reference to 'model drift' and 'warming drift' is confusing. Does this mean warming at depths below the AW core? Expansion of the AW core in depth? Please clarify.

Reply: We rewrote this part: "the thickening trend of the warm Atlantic Water layer (indicated by the deepening trend of the lower boundary of the warm Atlantic Water layer) remain large in four of the high-resolution models" L290-292

- Figure 7: I think some version of the PHC3.0 salinity profile should be included in the main paper, and I would recommend including Figure S1 as is.

Reply: now we put the salinity profiles in the main text and the plots of salinity biases in supplementary figures.

- Figure 12 and line 313-314: Are the observations also November to May averages?

Reply: yes. We revised the sentence to make it clear: "Fig. 12 depicts the MLD in winter (November to May) during the period 1979--2012 for each model and the observational estimates as well". L338-339

- Figure 13: Is this computed from averaged fields, or individual fields and then averaged?

Reply: in the previous version we computed the temporal mean temperature and then computed the depth. We changed this order now. The results are slightly different, but the main conclusion remains the same. We explained the calculation in Fig. 13 caption.

- Figure 14: Please clarify in the caption that positive values correspond to a deepening of the cold halocline depth (depth is positive downwards?).

Reply: We added in Fig. 14 caption: "A negative value indicates an uplift of the cold halocline base depth."

- Figure 3, 16: For basin averages, is the average taken over the entire area shown in e.g., Figure 15, or is a certain section of this region chosen?

Reply: We added the definition in the paper: "In this study, the Eurasian Basin and Amerasian Basin are defined as the deep ocean area with bottom topography deeper than 500~m and separated by the Lomonosov Ridge." L159-160

- Line 485: 'warming of the Pacific Water' by this you mean water in the Pacific Ocean and not Pacific Summer Water or Pacific Winter Water in the Arctic? (This manuscript doesn't actually show that the Pacific Ocean water is warming, only that the heat transport is increasing, a reference on this could be useful too).

Reply: it is the "warming of the Pacific Water inflow". We added references. L512

- Line 530: 'improves noticeably' should be 'increases noticeably and is likely an improvement'? We do not know what the actual interannual variability is? So it is difficult to say if the lower resolution of higher resolution is more accurate? Is this comment made assuming that high resolution models are closer to the 'truth'?

Reply: This sentence should be put at the end of that paragraph. According to the results discussed in that paragraph, the "variability" is improved in high-resolution models. L540

- Line 553-554: this statement about model spread should be stated earlier in the paper, as it is true for many of the variables considered here.

Reply: We state this earlier (in the section about Barents Sea Opening transports) now. L528-530

- Line 568-569: this would be good information to have in the introduction prior to reading through the results.

Reply: We moved it to the introduction section. L74-75

- Line 653 to 662: is some of the 'improvement' in interannual variability of gateway transports related to the improved simulation of eddies? Is there another cause for this change in interannual variability that should be speculated about?

Reply: We added a short discussion at the end of this paragraph: "Mesoscale eddies can influence the distribution of warm Atlantic Water between the inflow to the Arctic basin and the recirculation branch in Fram Strait (Wekerle2017, Hattermann2016). Some permitted eddies in the high-resolution models could contribute to the improvement in ocean heat inflow in Fram Strait in terms of mean state and variability. However, it has been suggested that 1 km resolution is needed to well simulate mesoscale eddies in Fram Strait and thus capture their effect (Wekerle2017). Hence, other factors, such as reduction in numerical mixing and better representation of the topographic steering of the flow with increasing resolution, should have played a more important role in changing the distribution of the Atlantic Water between its two branches and improving the inflow in Fram Strait." L677-683

**Reply to Reviewer #2**

Dear Reviewer,

Thank you very much for your very helpful comments. We revised the paper following your comments. See our reply below (in blue).

General comments

The topic of the study is very interesting and important. The ocean component of current coarse resolution climate models (CMIP6) has large biases in the Arctic Ocean, and future climate projections of the hydrography in the Arctic Ocean show a large model spread. Thus, it is necessary to investigate approaches that can reduce the model biases, such as using higher resolution versions of the ocean components. This is the focus of this study, to investigate how higher resolution versions improves the hydrography and ocean circulation in the Arctic Ocean compared to lower resolution versions. The authors use 5 different ocean components with realistic atmospheric forcing, where each component has a high and a low resolution version.

The manuscript is overall nicely written and has very interesting results with good figures. However, there are two major comments: (1) Some discussions points are missing in the present study, as listed below. Although discussion points, I think it is important to discuss some of these points early in the manuscript, perhaps as part of the approach used in the study (i.e., in Section 2). (2) The structure of Section 4 (Conclusion and Discussion) is to some extent difficult to follow. I therefore recommend major revision. I believe the authors will be able to improve on these two points. In addition, there are parts of the manuscript text that could be clarified and/or improved. I've pointed to some of these places further below (see 'Comments per section').

(1) The manuscript needs discussion:

- On how representative the 5 ocean models are compared to other CMIP6 ocean models (Shu et al., 2023). It seems that some of the 5 ocean models used in this study have poor performance compared to other CMIP6 ocean models.
- On the very different sets of resolution for each model. For instance, one ocean model has 9km as low resolution and 3.6km as high resolution, whereas another model has 72km as low resolution and 6.8km as high resolution.

Reply: we added discussions about "Representativeness of analyzed models" in the revised paper related to these aspects. L608-625

- On the impact of increased horizontal resolution when the vertical resolution is changing at the same time (in some cases the parameterization or sea ice model also changes when moving from low to high resolution version). This is to some extent discussed in Section 4.

Reply: in the revised version we have a separate subsection about horizontal resolution versus vertical resolution. L627-638

- On the deep-water balance, as the spin-up does not appear to be long enough.

Reply: we added discussions about spinup and initialization. L593-606

- On the benefits of high resolution version when the low resolution has already good performance.*
*The study shows that the high resolution versions outperforms the low resolution versions when the latter is showing poor performance (3 out of 5 models, where the low resolutions are 9km, 32km, and 51km). But when the low resolution has good performance, the benefits of the high resolution are not that clear (AWIFESOM). In addition, one model shows that higher resolution does not necessarily lead to improvements (IAP-LICOM). *The good performance of the low resolution of AWI-FESOM is mentioned in Section 4 (lines 610-612), and the discussion on lines 613-614 is very interesting. Could you please expand on this discussion; why is low resolution AWI-FESOM working quite well despite its coarse resolution?

Reply: The new subsection "Representativeness of analyzed models" is dedicated to discuss the related aspects. L608-625

(2) The structure and message of Section 4 needs to be clearer.

- Section 4 is, as it stands now, sometimes repetitive and needs more structure.

Reply: After splitting section 4 to two sections, we restructured them to avoid repetition and improve readability.

- The results could be more concrete; the results are sometimes a bit vague. For instance, how much is the performance improved by increasing the horizontal resolution? Could it be quantified?

Reply: In the revised conclusion section, we mentioned the quantitative improvement in temperature and salinity with increased resolution. Items 2-4.

- The results seem to be sometimes simplified, e.g., line 570. Could the discussion be more nuanced? For instance, one model shows good performance for both versions, one model shows overall poor results, whereas three models show poor results for low resolution but improved for the high resolution.

Reply: At this place we added "in four of the five models". L695. We also added some quantitative comparisons for temperature and salinity between the two sets of models as mentioned above.

- As the five different models show quite different results (and the configurations are different from model to model), it might be useful to discuss each of the models separately.

Reply: the individual models are discussed in the main text. In the final conclusion we would like to provide information on model ensemble comparisons and draw some general conclusions.

- I think it would benefit the manuscript to split Section 4 into two sections; first Discussion and thereafter summarize and conclude the study in Conclusion.

Reply: We followed your suggestion and split the section to two.

Comments per section

Abstract
Line 6: The main improvements appear to be seen in the Fram Strait and Davis Strait, but not much improvement seems to be the case in the Bering Strait. Suggest making the 'Arctic gateways' more nuanced.
Reply: we added "...at Fram and Davis straits". L6

Introduction
Line 15: Note that the amplified Arctic Ocean warming compared to global is for the future, or is the 'two times faster' referring to present?
Reply: The Arctic Ocean is already in the phase of amplified warming according to the cited reference.

Please clarify which models are ocean components forced with realistic atmosphere, coupled, regional or global models.
Reply: in the second paragraph of the introduction, we now wrote more explicitly about model types in previous intercomparison studies.

Results
Line 168-169: How does this relate to Heuze et al. (2023; this study is already in your reference list) that finds that the ocean heat transport through the Fram Strait is reasonable, but the volume transport and temperature is not correct?
Reply: in the Fram Strait transport section (3.5.3) we show that the heat transports are underestimated in these low-resolution models. So this is different to Heuze et al. 2023.

Line 175: The recent study by Richards et al. (2022) might be useful here.
Reply: we added the citation. L192

Line 172-180: Here the boundary currents are discussed. Have you looked into the ocean circulation/boundary currents in the models?

Reply: the relatively high temperature of Atlantic-origin water along the boundary is a good indicator of the boundary current. We did not explore velocity fields. We revised the text to be clear that we are referring to the Atlantic Water spatial pattern as shown in the figure rather than the current. L189-192

Line 184-185: This sentence needs a reference. Could there be cold sources in the deep also entering through the Fram Strait, such as Norwegian Sea Deep Water?
Reply: we added citations. According to literature, the "main" supplier of the deep water is dense shelf waters. L202-204

Line 192: The results described here are difficult to see from the figures.
Reply: the sentence is removed.

Line 198: It is not fully clear how the absence of the sea ice dynamics impacts the results.
Reply: Missing sea ice dynamical process could lead to unrealistic sea ice and ocean surface fluxes. We added this in the sentence. L211-212

Line 219: The discussion on how the sea surface salinity restoring impacts the results is not fully clear to me.
Rely: The results basically tell that the models require applying numerical restoring to sea surface salinity climatology to maintain a good simulation of salinity. We have a short discussion in the discussion section about this: "Although salinity biases in the halocline are reduced in some high-resolution configurations, two models exhibit similar or even larger sea surface salinity biases when their spatial resolution is improved. This could potentially be attributed to weak or absent sea surface salinity restoring. The strong dependence of simulated sea surface salinity on numerical restoring indicates the general need for improvements in the surface freshwater budget and the processes influencing freshwater circulation and distribution, such as the impact of sea ice on momentum transfer in models.". L657-662

Line 225: A reference is needed.
Reply: Added. L240

Line 254: Poleward propagation of warm anomalies in the Atlantic Water layer is discussed in Årthun et al. (2017).
Reply: We add this reference at a more adequate place (L470).

Line 259: How is sea ice decline leading to intensified inflow?
Reply: we added the explanation: "During the 2010s, both the warming and intensification of the inflow in the Fram Strait due to Arctic sea ice decline, which maintained the strength of the cyclonic Greenland Sea gyre circulation by reducing sea ice freshwater export through Fram Strait, contributed to the warming of the Atlantic Water layer in the Arctic basin" L278-281

Line 260: Please be consistent, use either Arctic basin or Arctic Basin.
Reply: we use "Arctic basin" in all cases.

Line 270-272: This is not so easy to see from the figures.
Reply: the sentence is removed.

Line 280: Please explain why these two periods are compared.
Reply: we add that this is the "available" data set. L300

Line 349: Note that a different approach is taken in Muilwijk et al. (2023) studying the halocline in CMIP6 models.
Reply: we simply employ the available halocline base depth observations to evaluate the OMIP models and do not consider more approaches for a reasonable paper length.

Line 363: Is it the deepening of the Atlantic Water layer that causes the deepening of the halocline, or opposite, or other factors causing the deepening of the layers?
Reply: the sentence was incorrect. It is changed to "The anomaly of the cold halocline base depth in the Amerasian Basin in IAP-LICOM is not consistent with the observations". L386-387

Line 368: Should I expect to see that 1990s is warmer than 2010s for FSU-HYCOM? Please clarify.
Reply: It is rather the case in CMCC-NEMO. We removed FSU-HYCOM here.

Line 417: A positive salinity bias both in the Amerasian Basin and the Eurasian Basin?
Reply: There is positive bias in both basins, but the salinity trend over the 40 years is dominated by the Amerasian Basin.

Conclusion and Discussion

Please see the comments above regarding Section 4.
Several places in this section it is mentioned that higher resolution helps, but at the same time other findings reveal that the answer is not that straightforward (for instance AWI-FESOM and IAP-LICOM show different results than the three other models). This comment is similar to my comment above, about giving a bit more nuanced discussion.
Reply: We followed your suggestion to split the section and restructure and revise the text. In the conclusion section, we do not mention model names (instead, we say, for example, four out of five models); in the discussion section we mention model names when needed to be clearer.

Line 629: The results from Pan et al. (2023) are relevant here.
Reply: We added a reference to Shu et al. 2019 here, which is about the relationship of model biases in the Arctic basin and shelf seas, being more relevant than Pan et al. 2023. L651

Figures
Fig. 14: The dark red color in figure 14j is very difficult to see. Perhaps write a note in the caption that the Amerasian Basin is deep red in 14j.
Reply: we added the sentence.

References:

Årthun, M., Eldevik, T., Viste, E. et al. Skillful prediction of northern climate provided by the ocean. Nat Commun 8, 15875 (2017). https://doi.org/10.1038/ncomms15875

Richards, A. E., Johnson, H. L., & Lique, C. (2022). Spatial and temporal variability of Atlantic Water in the Arctic from 40 years of observations. Journal of Geophysical Research: Oceans, 127, e2021JC018358. https://doi.org/10.1029/2021JC018358

---

## Author Response (AR2)

Dear reviewer,
Thank you for your suggestions. Below are our replies.

Table 1: I'm not overly familiar with sea surface salinity restoring, a footnote on units would be useful.
Reply: a footnote is added

Line 159-160: averaging over the Eurasian basin, does this includes the Greenland – Iceland - Norwegian seas?
Reply: We added "In this study, the Arctic Ocean is defined as the Arctic area enclosed by Fram Strait, Barents Sea Opening, Bering Strait and the northern boundary of Canadian Arctic Archipelago, and the Eurasian Basin and Amerasian Basin are defined as the deep Arctic Ocean areas with bottom topography deeper than 500 m and separated by the Lomonosov Ridge." L151-153

Figure 12 and 13: blue colors representing shallow depths is not intuitive to me. Consider flipping the colorbar or using a different colorbar for these figures.
Reply: colormap is changed from a divergent one to a sequential one.

Line 427-428: consider rephrasing to avoid overuse of negatives (reduction in dissimilarity is not pronounced).
Reply: changed to "With the increase in model resolution, the consistency of the simulated freshwater column among the models is not clearly improved." L429-430.

Best regards
Qiang